# Conservation of peripheral nervous system formation mechanisms in divergent ascidian embryos

Joshua F Coulcher[1†], Agnès Roure[1†], Rafath Chowdhury[1], Méryl Robert[1], Laury Lescat[1‡], Aurélie Bouin[1§], Juliana Carvajal Cadavid[1], Hiroki Nishida[2], Sébastien Darras[1]*

[1]Sorbonne Université, CNRS, Biologie Intégrative des Organismes Marins (BIOM), Banyuls-sur-Mer, France; [2]Department of Biological Sciences, Graduate School of Science, Osaka University, Toyonaka, Japan

**Abstract** Ascidians with very similar embryos but highly divergent genomes are thought to have undergone extensive developmental system drift. We compared, in four species (*Ciona* and *Phallusia* for Phlebobranchia, *Molgula* and *Halocynthia* for Stolidobranchia), gene expression and gene regulation for a network of six transcription factors regulating peripheral nervous system (PNS) formation in *Ciona*. All genes, but one in *Molgula*, were expressed in the PNS with some differences correlating with phylogenetic distance. Cross-species transgenesis indicated strong levels of conservation, except in *Molgula*, in gene regulation despite lack of sequence conservation of the enhancers. Developmental system drift in ascidians is thus higher for gene regulation than for gene expression and is impacted not only by phylogenetic distance, but also in a clade-specific manner and unevenly within a network. Finally, considering that *Molgula* is divergent in our analyses, this suggests deep conservation of developmental mechanisms in ascidians after 390 My of separate evolution.

***For correspondence:**
sebastien.darras@obs-banyuls.fr

[†]These authors contributed equally to this work

**Present address:** [‡]Department of Developmental and Molecular Biology, Albert Einstein College of Medicine, New York, United States; [§]Toulouse Biotechnology Institute, Université de Toulouse, CNRS, INRAE, INSA, Toulouse, France

**Competing interests:** The authors declare that no competing interests exist.

## Introduction

The formation of an animal during embryonic development is controlled by the exquisitely precise deployment of regulator and differentiation genes in space and time. This coordinated expression of developmental genes is controlled through gene regulatory networks (GRNs). It has been proposed that these networks are encoded in the genome and control body plan, organs, and cell type formation (*Levine and Davidson, 2005*). It is thought that, in the course of evolution, essential GRNs or sub-modules have been conserved and control the formation of homologous structures, and that rewiring of the GRNs explains acquisition of novelties and morphological diversification (*Davidson and Erwin, 2006*). Understanding control of gene expression is central to comparative developmental approaches. Hence, specific domains of the non-coding genome called *cis*-regulatory modules (CRMs) that serve as docking platforms for transcription factors (TFs) (*Buffry et al., 2016*; *Rebeiz and Tsiantis, 2017*) have been deeply studied. It is generally considered that during evolution and species diversification the key functional role of CRMs leads to their persistence and indirectly contributes to maintaining genes together (synteny) and to the existence of conserved non-coding DNA. Indeed, the presence of TF-binding sites (TFBS) in the CRMs for genes with conserved expression and regulation defines regions of conserved sequence; and searching for conservation in non-coding DNA is often used to identify CRMs (phylogenetic footprint: defined as DNA sequence conservation between related species following alignment using tools such as lastz or lagan [*Brudno et al., 2003*; *Schwartz et al., 2003*]). Such conserved CRMs have been identified in closely

related species but also to a fewer extent for very distantly related species (ultraconserved elements); and in many cases, gene regulation relies on the usage of a similar combination of TFBS.

By contrast, a number of comparative studies performed in several group of animals (chordates, insects, nematodes...) have revealed that development is flexible and that formation of homologous characters may rely on different mechanisms, a process known as developmental system drift (DSD) (*Buffry et al., 2016*; *Arnold et al., 2014*; *Verster et al., 2014*; *True and Haag, 2001*; *Haag, 2014*; *Bradley et al., 2010*; *José-Edwards et al., 2015*; *Lusk and Eisen, 2010*; *Roure et al., 2014*). In some cases, key GRN nodes have changed (for example the role of *Snail/Slug* genes in vertebrate neural crest [*Nieto, 2018*]). Focusing on GRN edges — gene expression regulation — various situations have been described, and conservation of CRM sequence at the DNA level does not necessarily correlate with conservation of activity. First, conservation of DNA sequence across species, does not imply a *cis*-regulatory activity (for example, ultraconserved elements do not necessarily function as CRMs or may be dispensable for gene expression) or an identical regulatory logic (conserved activity may not involve the same TFBS; conserved CRMs may control different expression patterns depending on the species). On the contrary, DNA sequence conservation is not a necessary feature of CRMs. First, bona fide CRMs may not be conserved, even between closely related species. Second, orthologous CRMs with conserved activity harboring similar combinations of TFBS exist despite an absence of sequence conservation because of extensive turnover of TFBS.

Ascidians are marine invertebrates that are particularly interesting for investigating the evolution of gene regulation during embryonic development. They belong to the tunicate phylum, and together with vertebrates and cephalochordates (amphioxus) they constitute the chordate superphylum (*Lemaire, 2011*; *Satoh, 1994*; *Satoh et al., 2014*). These animals share a body plan characterized by a notochord and a dorsal neural tube during embryonic life. Ascidians, however, took a significantly different evolutionary path from other chordates resulting in divergent morphological, embryological, and genomic features. The development of these animals is fast (hatching of the larva after 18 hr at 18°C in *Ciona robusta* [*Hotta et al., 2007*]) and stereotyped with very few cells (around 100 cells in gastrulae and 2500 cells in larvae [*Nishida, 1986*; *Yamada and Nishida, 1999*]), and ascidian genomes have undergone compaction and extensive rearrangements when compared to vertebrates (*Dehal et al., 2002*; *Brozovic et al., 2017*). In addition, ascidians have extensively diversified (around 3000 species) and their genomes have been reshuffled to the point that there is very little DNA conservation outside of the coding parts of the genome (except for closely related species such as the ones from the same genus) (*Shenkar and Swalla, 2011*; *Dardaillon et al., 2020*). Yet, their embryos are virtually identical. This paradoxical situation represents an excellent case to study the evolution and diversification of developmental mechanisms, CRMs and DSD (*José-Edwards et al., 2015*; *Roure et al., 2014*; *Stolfi et al., 2014*; *Madgwick et al., 2019*; *Colgan et al., 2019*; *Takahashi et al., 1999*; *Hudson et al., 2011*; *Racioppi et al., 2017*; *Johnson et al., 2004*; *Brown et al., 2007*; *Oda-Ishii et al., 2005*). Ascidians are simple tractable organisms and are excellent models for functional genomics, in particular with plasmid DNA electroporation that allows the easy generation of thousands of transient transgenic embryos. Moreover, several species at various phylogenetic distances with sequenced and annotated genome are amenable to experimentation (*Dardaillon et al., 2020*; *Kourakis and Smith, 2015*).

Several in details developmental studies have already been performed in ascidians, and the comparative conclusions are contrasted (reviewed in *Lemaire, 2011*; *Kourakis and Smith, 2015*; *Hudson and Yasuo, 2008*; *Lemaire et al., 2008*; *Lemaire and biology, 2006*; *Satoh, 2014* *Razy-Krajka and Stolfi, 2019*). First, comparisons with vertebrates suggest that the 550 My of separate evolution has led to significant differences in the early specification of major chordate tissues, such as the notochord or the central nervous system. But deep similarities have been uncovered in the GRN controlling heart or placode formation for example. Second, comparisons between distantly related ascidian species suggest that although the embryological and cellular processes are highly similar, the developmental regulators (TFs and signaling molecules) may change. Third, while conservation of developmental gene expression has been reported even at large phylogenetic distance, the mechanisms for such conservation relies on shared regulatory mechanism (i.e. conserved but shuffled TFBS in possibly non-alignable CRMs), or by contrast on different mechanisms (extensive DSD).

Comparative studies within ascidians are largely based on data from the most studied species *Ciona intestinalis* that have been compared with equivalent results obtained from another species

belonging to a different family. Importantly, the estimated divergence times with *C. intestinalis* are considerable (*Delsuc et al., 2018*): 110 My for *Ciona savignyi*, 275 My for *Phallusia mammillata* or *Corella inflata* and 390 My for *Halocynthia roretzi* or *Molgula* species.

Here, we have tried to expand the phylogenetic range by comparing gene expression regulation in several ascidian species, mainly *C. intestinalis*, *P. mammillata* and *Molgula appendiculata*. We have studied a subset of the GRN controlling caudal peripheral nervous system (PNS) formation. Caudal PNS formation is well understood in *C. intestinalis* both at the level of the developmental and cellular processes, and at the level of the molecular regulators (*Roure et al., 2014*; *Pasini et al., 2006*; *Pasini et al., 2012*; *Roure and Darras, 2016*; *Feinberg et al., 2019*; *Waki et al., 2015*; *Chen et al., 2011*; *Joyce Tang et al., 2013*; *Horie et al., 2008*; *Candiani et al., 2005*; *Torrence and Cloney, 1982*). First, dorsal and ventral epidermis midlines are induced as neurogenic territories by Fgf and Bmp signals from the underlying endomesodermal cells, respectively. Then, the number of glutamatergic PNS neurons (caudal epidermal sensory neurons or CESNs) that form in these neurogenic midlines is controlled by the Notch pathway. Molecular markers including a number of developmental regulators have been described for these two key phases of caudal PNS formation. However, their functions and direct interactions have been determined for only a limited number (*Figure 1—figure supplement 1A*). Focusing on seven midline TFs, we identified CRMs that are active in neurogenic midlines from both *C. intestinalis* and *P. mammillata*. Reciprocal cross-species transcriptional assays pointed to a strong conservation of the *cis*-regulatory logic between these species despite a lack of sequence conservation in these CRMs. Conserved activity was also observed in two additional species that both belong to the Phlebobranchia order. However, conserved activity was observed for only a minority of CRMs for some nodes of this GRN in *M. appendiculata* that belongs to the Solidobranchia order. Surprisingly, CRMs isolated from another Solidobranchia, *Halocynthia roretzi*, did show remarkable activity in the *P. mammillata* midlines but not in *M. appendiculata*. Overall, our results suggest deep conservation of gene regulation in caudal PNS formation within ascidians and advocate for broad phylogenetic sampling in comparative studies.

## Results

### Identification of midline CRMs in *C. intestinalis*

As described above, caudal PNS formation is initiated by two inductive events: Fgf9/16/20 as the inducer of the dorsal neurogenic midline (DML) and Admp as the inducer of the ventral neurogenic midline (VML). While function and regulatory interactions have been determined for several genes, direct interactions, and associated CRMs are unknown for the most part (*Figure 1—figure supplement 1A*; *Roure et al., 2014*; *Pasini et al., 2006*; *Roure and Darras, 2016*; *Waki et al., 2015*; *Joyce Tang et al., 2013*; *Bertrand et al., 2003*). Here, we have focused our work on seven TFs that are expressed in both ventral and dorsal caudal neurogenic midlines (VDML) and for which we have already described regulatory interactions that generated a provisional GRN (*Pasini et al., 2006*; *Roure and Darras, 2016*). These genes are — ordered by their onset of expression in VDML — *Msx*, *Ascl.b*, *Klf1/2/4*, *Nkxtun3*, *Tox*, *Dlx.c* and *Bhlhtun1*. *Msx*, *Nkxtun3* and *Dlx.c* code for homeodomain-containing TFs; *Ascl.b* and *Bhlhtun1* for bHLH TFs; *Klf1/2/4* for a C2H2 Zn finger TF; and *Tox* for a HMG TF. We aimed at identifying regions of genomic DNA that behave as midline CRMs. Previous work in the ascidian community has shown that selecting a few kb in conserved non-coding DNA generally upstream of the gene of interest had high chance of identifying most, if not all, regulatory regions. We thus selected upstream regions of up to 7 kb containing phylogenetic footprints between *C. robusta* and *C. savignyi* that we placed upstream of the minimal promoter of the *Ciinte. Fog* gene and *LacZ* as a reporter, and tested in vivo before further reduction in size. After our work was performed, open chromatin regions from ATAC-seq data were identified from whole embryos up to neurula stages (*Madgwick et al., 2019*) and it turned out that most of the midline CRMs that we identified correspond to conserved regions with enrichment in ATAC-seq (*Figures 1*, *2*, *3*, *Supplementary file 1* and *2*). Tissue-specific ATAC-seq data should thus have a high predictive power and limit the time-consuming promoter-bashing approach in future experiments seeking for active CRMs.

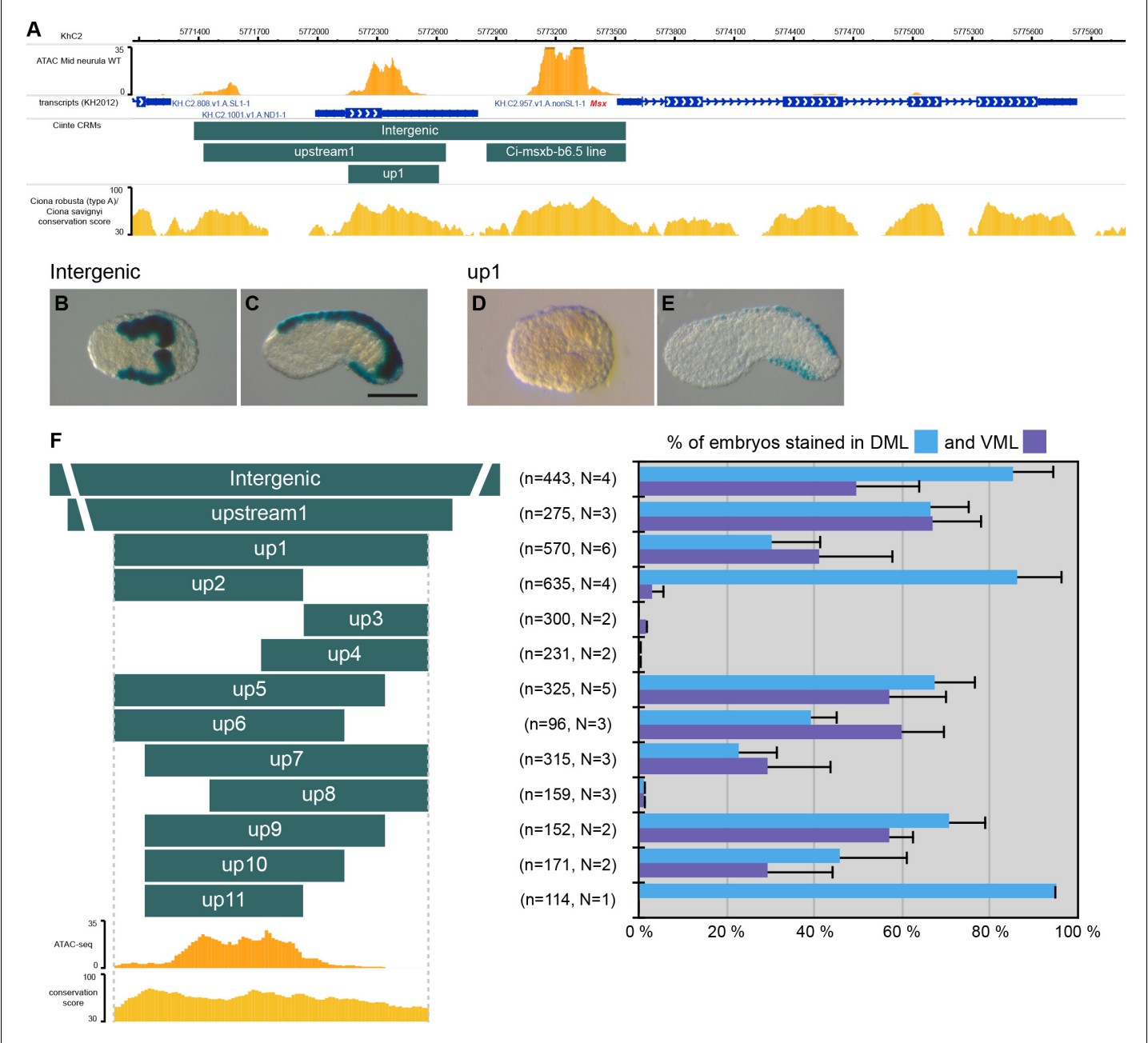

**Figure 1.** CRMs controlling *Ciinte.Msx* expression in VDML. (**A**) Snapshot of the *Ciinte.Msx* locus depicting ATAC-seq profile at mid-neurula stages, tested genomic regions, transcript models and conservation between *C. robusta* and *C. savignyi* (from https://www.aniseed.cnrs.fr/ and *Dardaillon et al., 2020*; *Madgwick et al., 2019*). (**B–E**) Representative examples of X-gal stained embryos at late gastrula stages (**B, D**) and early tailbud stages (**C, E**) following *C. intestinalis* embryos electroporation of Ciinte.Msx-Intergenic (**B, C**) and Ciinte.Msx-up1 (**D, E**). Embryos are shown in dorsal view (**B, D**) and in lateral view with dorsal to the top (**C, E**), and anterior to the left. Scale bar: 100 μm. (**F**) Schematic representation of the various constructs and their activity at early tailbud stages in DML (blue) and VML (purple) (n indicates the total number of embryos examined; N indicates the number of independent experiments).

The online version of this article includes the following figure supplement(s) for figure 1:

**Figure supplement 1.** Developmental regulators of caudal PNS in *C. intestinalis*.

**Figure supplement 2.** Model for *Ciinte.Msx* regulation.

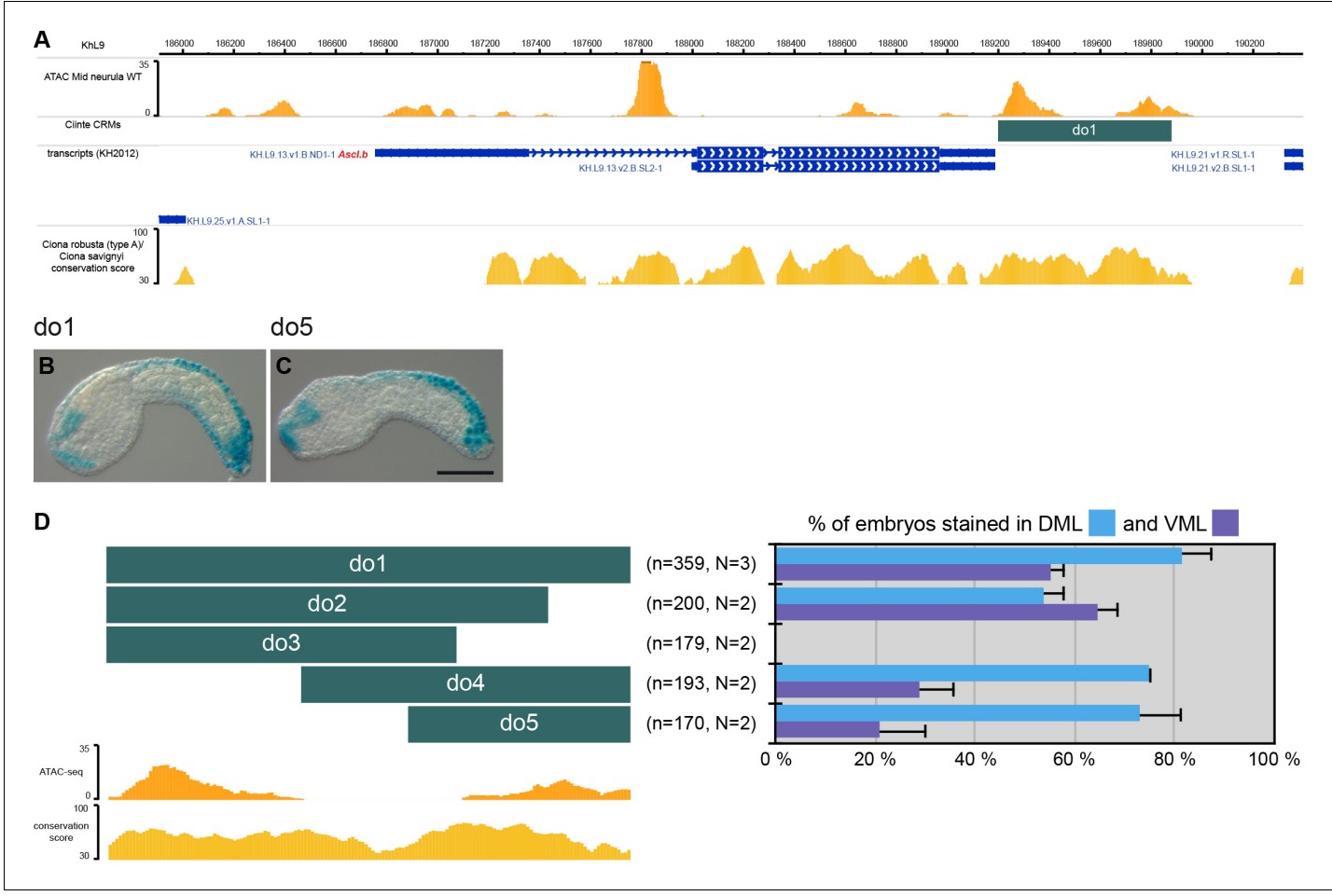

**Figure 2.** CRMs controlling *Ciinte.Ascl.b* expression in VDML. (**A**) Snapshot of the *Ciinte.Ascl.b* locus depicting ATAC-seq profile at mid-neurula stages, tested genomic regions, transcript models and conservation between *C. robusta* and *C. savignyi* (from https://www.aniseed.cnrs.fr/ and *Dardaillon et al., 2020*; *Madgwick et al., 2019*). (**B–C**) Representative examples of X-gal stained embryos at early tailbud stages following *C. intestinalis* embryos electroporation of Ciinte.Ascl.b-do1 (**B**) And Ciinte.Ascl.b-do5 (**C**). Embryos are shown in lateral view with dorsal to the top and anterior to the left. Scale bar: 100 μm. (**D**) Schematic representation of the various constructs and their activity at early tailbud stages in DML (blue) and VML (purple) (n indicates the total number of embryos examined; N indicates the number of independent experiments).

The online version of this article includes the following figure supplement(s) for figure 2:

**Figure supplement 1.** Putative TFBS.

## Uncoupling of dorsal and ventral activities for the early genes

### Msx

*Msx* is expressed in DML precursors from the 64-cell stage while it is expressed in the VML from the gastrula/neurula stages (*Roure and Darras, 2016*). We have previously described a Nodal/Otx-regulated CRM located immediately upstream of the gene responsible for the early expression in DML precursors (Ci-msxb-b6.5 line in *Figure 1A*; *Roure et al., 2014*). Here, we show that the entire upstream intergenic region is active early in the DML (*Figure 1B*) but also later in the VML (*Figure 1C*). A distal fragment, up1, adjacent to the early CRM was active in both VML and DML at tailbud stages but inactive at gastrula/neurula stages (*Figure 1D–F*). These results show that the seemingly continuous expression of *Msx* in the DML is in fact under the control of two separate CRMs that are likely regulated differently (see below). We next generated various deletions of this VDML CRM (*Figure 1F* and *Supplementary file 2*). We could observe VDML activity for most regions, with generally a decrease of strength when the size of the fragment was reduced, or a lack of activity. The smallest fragment active in VDML, up10, was 291 bp long. Interestingly, two overlapping regions, up2 and up11, were active only in the DML. Consequently, even if we did not identify

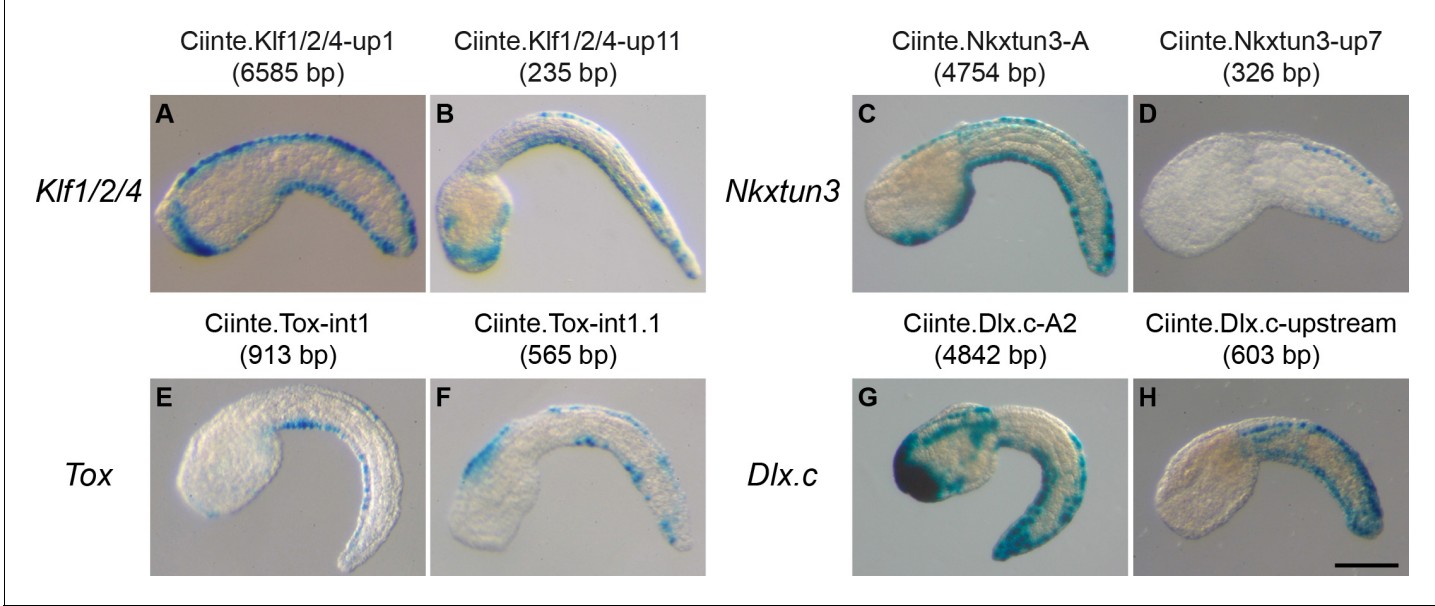

**Figure 3.** CRMs with activity in the tail epidermis midlines for *Ciinte.Klf1/2/4*, *Ciinte.Nkxtun3*, *Ciinte.Tox*, and *Ciinte.Dlx.c*. Representatives examples of X-gal stained embryos at tailbud stages following *C. intestinalis* embryos electroporation of *C. intestinalis* genomic regions for *Klf1/2/4* (A, B), *Nkxtun3* (C, D), *Tox* (E, F) and *Dlx.c* (G, H). For each gene, an example for the largest and the smallest regions with robust VDML activity are shown (the size of the region is shown between parentheses after the CRM's name). Embryos are shown in lateral view with dorsal to the top and anterior to the left. Scale bar: 100 μm.

The online version of this article includes the following figure supplement(s) for figure 3:

**Figure supplement 1.** CRMs controlling *Ciinte.Klf1/2/4* expression in VDML.
**Figure supplement 2.** CRMs controlling *Ciinte.Nkxtun3* expression in VDML.
**Figure supplement 3.** CRMs controlling *Ciinte.Tox* expression in VDML.
**Figure supplement 4.** CRMs controlling *Ciinte.Dlx.c* expression in VDML.

a CRM specific for the VML, this deletion analysis suggests that ventral and dorsal expressions of *Msx* are uncoupled, a conclusion in agreement with the proposed role of *Msx* as an upstream regulator of the neurogenic GRN and an integrator of dorsal and ventral inducing cues (***Roure and Darras, 2016***). This scenario is further supported by the presence of conserved TFBS for candidate ventral and dorsal regulators (***Figure 1—figure supplements 1–2***; Material and methods). Ciinte.Msx-up10 contains two putative Msx-binding sites that could account for activation in the DML. Activation by Bmp signaling in the VML could be direct (4 Bmp-Responsive Elements (BRE) and 3 Smad-Binding Elements (SBE)) or indirect (sites for the ventral TFs Irx.c and Tbx2/3).

### Ascl.b
Although the upstream intergenic region (0.8 kb) was not active (***Supplementary file 2***), the 0.7 kb region immediately downstream of the gene, do1, was robustly active in VDML (***Figure 2*** and ***Supplementary file 2***). Fragments of this region were either not active, active in VDML or mainly active in the DML. Consequently, similarly to *Msx*, the VML and DML expression of *Ascl.b* appear to be distinctly regulated. While we identified putative sites for dorsal and ventral factors in the do2 fragment (***Figure 2—figure supplement 1***), they are not sufficient to explain VDML activity. This suggests that *Ascl.b* expression relies on additional unidentified factors.

### Pan-midline CRMs for late midline genes
For the five other TF genes, we could also isolate genomic regions active in VDML, except for *Bhlhtun1* for which the three tested regions were active in endogenous expression territories (palps, notochord...) but not in VDML (***Supplementary file 2*** and ***3***).

### Klf1/2/4

A 6.6 kb region, up1, immediately upstream of the gene was strongly active in VDML (*Figure 3A* and *Figure 3—figure supplement 1*). Various deletions were generated; most were active in VDML but with a lower efficiency (*Figure 3—figure supplement 1*). Analysis of putative TFBS suggested activation by Msx and Ascl.b (*Figure 3—figure supplement 1*).

### Nkxtun3

A 4.8 kb region, A, immediately upstream of the gene was strongly active in VDML (*Figure 3C* and *Figure 3—figure supplement 2*). Various deletions were generated; most were active in VDML but with a lower efficiency (*Figure 3—figure supplement 2*). We could identify a 326 bp fragment, up7, which was robustly active in VDML (*Figure 3D*). It is located 0.7 kb upstream of *Nkxtun3* and corresponds to a conserved region with an ATAC-seq enrichment detected from late gastrula stages (*Supplementary file 2*). TFBS analysis suggested activation by the upstream factors Msx and Klf1/2/4 (*Figure 3—figure supplement 2*).

### Tox

*Tox* is mainly transcribed as two isoforms that vary in their transcription start sites and first exons (*Figure 3—figure supplement 3* and *Supplementary file 2*; *Roure and Darras, 2016*). We selected a 913 bp conserved region, int1, located upstream of one isoform and in the first intron of the other. This region is active in VDML (*Figure 3E*). Two shorter regions were generated: int1.1 (565 bp) that is more active (*Figure 3F*) and int1.2 (138 bp) that is less (*Figure 3—figure supplement 3*). Putative binding sites analysis suggested activation by the upstream factors Ascl.b and Klf1/2/4.

### Dlx.c

A 4.8 kb region, A2, immediately upstream of the gene was strongly active in VDML (*Figure 3G* and *Figure 3—figure supplement 4*). Various deletions were generated; most were active in VDML but with a lower efficiency (*Figure 3—figure supplement 4*). The region 'upstream' (604 bp), located 1.3 kb upstream of *Dlx.c*, contains two blocks with higher sequence conservation and ATAC-seq enrichment, and was robustly active (*Figure 3H*). up7 (206 bp) containing the proximal block was active in VDML albeit at a low level (*Figure 3—figure supplement 4*). It does not contain any site for the putative upstream VDML regulators (Msx, Ascl.b, Klf1/2/4, and Nkxtun3), suggesting the implication of unidentified factor(s).

Interestingly, for all these four TFs coding genes (*Klf1/2/4*, *Nkxtun3*, *Tox*, and *Dlx.c*), starting from relatively large regions (up to 6.6 kb), we could reduce the size down to 200–300 bp and maintain VDML activity (*Figure 3*). Shortening the DNA regions was usually accompanied by a decrease in activity. However, all tested regions, despite some variability, had similar activity in both VML and DML; and we conclude that the expression of these genes is regulated by 'pan-midline' CRMs in agreement with their downstream positions in the PNS GRN (*Roure and Darras, 2016*).

Overall, results from promoter bashing in *C. intestinalis* led to three main conclusions: genomic support for VDML expression has been identified for most midline TFs (and this will be used to probe gene regulation conservation in other species); *Msx* and *Ascl.b* have been confirmed as upstream regulators of the caudal PNS GRN since dorsal and ventral expression are differently regulated; and downstream genes are regulated similarly dorsally and ventrally.

## Tail PNS specification appears to be conserved in *P. mammillata*

We next wondered whether caudal PNS was specified by similar mechanisms in *P. mammillata*, an ascidian species that diverged from *C. intestinalis* around 275 My ago (*Delsuc et al., 2018*; *Figure 4*). We have previously shown that the early induction of DML fate by Fgf/Nodal signals is conserved and leads to the expression of the early DML precursors markers *Msx* and *Dlk* (previously known as *Delta2*) (*Roure et al., 2014*).

### Expression and regulation of caudal PNS genes

By in situ hybridization, we first determined the expression pattern of orthologous genes for the caudal PNS genes (for each gene, we could identify a single ortholog in the *Phallusia* genome). Importantly, we found that the seven neurogenic TFs, which are the focus of the present study, were

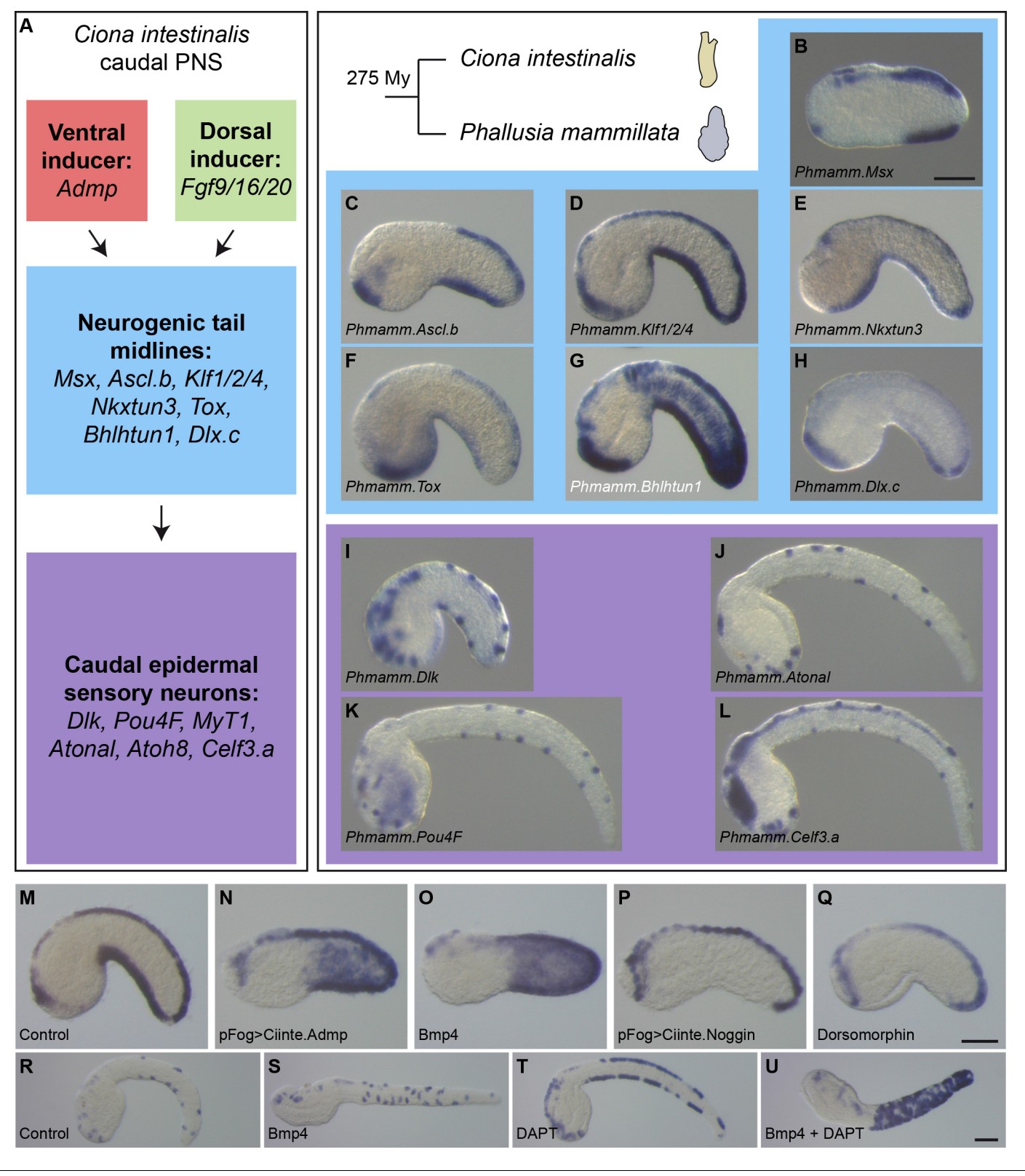

**Figure 4.** Expression and regulation of caudal PNS genes in *Phallusia mammillata*. (**A**) Schematic summary of caudal PNS specification and molecular regulators in *Ciona intestinalis* (adapted from ***Roure et al., 2014***; ***Pasini et al., 2006***; ***Roure and Darras, 2016***; ***Waki et al., 2015***; ***Joyce Tang et al., 2013***, more details can be found in ***Figure 1—figure supplement 1A***). (**B–L**) Expression of the *P. mammillata* orthologs of the *C. intestinalis* caudal

*Figure 4 continued on next page*

*Figure 4 continued*

PNS genes. In situ hybridization of a late neurula for *Msx* (B), at early tailbud stages for *Ascl.b* (C), *Klf1/2/4* (D), *Nkxtun3* (E), *Tox* (F), *Bhlhtun1* (G), *Dlx.c* (H) and *Dlk* (I); and at mid/late tailbud stages for *Atonal* (J), *Pou4F* (K), and *Celf3.a* (L). (M–Q) Expression of *Phmamm-Klf1/2/4* at early tailbud stages in control embryos (M), following electroporation of pFog >Ciinte.Admp (N) or pFog >Ciinte.Noggin (P); or following treatment with Bmp4 protein (O) or Dorsomorphin (Q). (R–U) Expression of *Phmamm.Pou4F* at late mid/late tailbud stages in control embryos (R), following treatment with Bmp4 protein (S), DAPT (T) or a combination of Bmp4 and DAPT (U). Embryos are shown in lateral view with dorsal to the top and anterior to the left. Scale bars: 50 μm.

The online version of this article includes the following figure supplement(s) for figure 4:

**Figure supplement 1.** Expression pattern of caudal PNS genes in *Phallusia mammillata*.

expressed in VDML like in *Ciona* and with apparently the same temporal sequence (**Figure 4B–H** and **Figure 4—figure supplement 1**). Similarly, the neuronal genes *Dlk*, *Atonal*, *Pou4F*, and *Celf3.a* were also expressed as spaced spots in VDML, likely corresponding to CESNs (**Figure 4I–L** and **Figure 4—figure supplement 1**). Interestingly, other sites of expression (palps, notochord, trunk neurons, central nervous system…) were also conserved (**Figure 8—figure supplement 3**).

We next determined whether Bmp and Notch signaling pathways could regulate caudal PNS formation. Treatment of whole embryos with zebrafish recombinant Bmp4 protein from the eight-cell stage led to ectopic expression in the lateral tail epidermis of *Msx* (BSA control: 20/20 embryos in VDML, N = 1; 150 ng/ml Bmp4: 27/27 embryos with ectopic expression, N = 1), *Klf1/2/4* (BSA control: 100% embryos in VDML, N = 9, n = 172; 150 ng/ml Bmp4: 100% embryos with ectopic expression, N = 6, n = 127; **Figure 4M,O**), and *Pou4F* (BSA control: 100% embryos in VDML, N = 4, n = 101; 150 ng/ml Bmp4: 94% embryos with ectopic expression, N = 4, n = 95; **Figure 4R,S**). Overexpression of the *Ciona* Bmp ligand Admp in the ectoderm by electroporation led also to ectopic *Klf1/2/4* expression (pFog >Venus control: 100% embryos in VDML, N = 4, n = 75; pFog >Ciinte.Admp: 71% embryos with ectopic expression, N = 1, n = 14; **Figure 4M,N**). Bmp inhibition using the pharmacological inhibitor Dorsomorphin had variable effects depending on the experiment, but we observed, as in *Ciona*, repression of *Klf1/2/4* expression in the VML (BSA control: 100% embryos in VDML, N = 9, n = 172; 20 μM Dorsomorphin: 5% to 70% embryos with VML repression with an average of 37%, N = 9, n = 200; **Figure 4M,Q**). Similarly overexpression of the secreted Bmp antagonist Noggin from *Ciona* led to VML repression of *Klf1/2/4* expression (pFog >Venus control: 100% embryos in VDML, N = 4, n = 75; pFog >Ciinte.Noggin: 0% to 80% embryos with VML repression with an average of 29%, N = 4, n = 109; **Figure 4M,P**). Inhibition of Notch signaling with the pharmacological inhibitor DAPT led to a massive increase of the *Pou4F*-positive cells in VDML; and when combined with Bmp4 treatment, the cells were found throughout the tail epidermis (BSA control: 100% embryos in VDML, N = 4, n = 101; 25 μM DAPT: 99% embryos with ectopic expression in VDML, N = 3, n = 94; 25 μM DAPT + 150 ng/ml Bmp4: 93% embryos with ectopic expression, N = 4, n = 118; **Figure 4R,T,U**).

## Identification of midline CRMs

To define VDML CRMs in *P. mammillata*, we undertook the same approach as the one performed in *C. intestinalis* but with a more limited generation of deletion constructs. We selected regions containing blocks of sequence conservation between *P. mammillata* and *P. fumigata* that we placed upstream of the minimal promoter for the *Ciinte.Fog* gene and *LacZ* as a reporter. Out of the seven targeted genes, we were unable to isolate CRMs with VDML activity for *Ascl.b* and *Bhlhtun1* (**Supplementary file 4** and **5**).

## Msx

Similarly to the situation in *C. intestinalis*, we have previously isolated Pm-msxb-b6.5 line, a CRM immediately upstream of *Msx* that is active early in the DML precursors (b6.5 lineage) (**Roure et al., 2014**). The upstream intergenic region that contains this CRM was active at tailbud stages in VDML (**Figure 5A** and **Figure 5—figure supplement 1**), and allowed to define a distal CRM, LED, that was active in VDML and corresponds to a conserved region with a weak ATAC-seq signal (**Figure 5—figure supplement 1** and **Supplementary file 5**). The topology of the *Msx* locus is strikingly similar between *C. intestinalis* and *P. mammillata*: there is local synteny (the gene upstream of *Msx* is

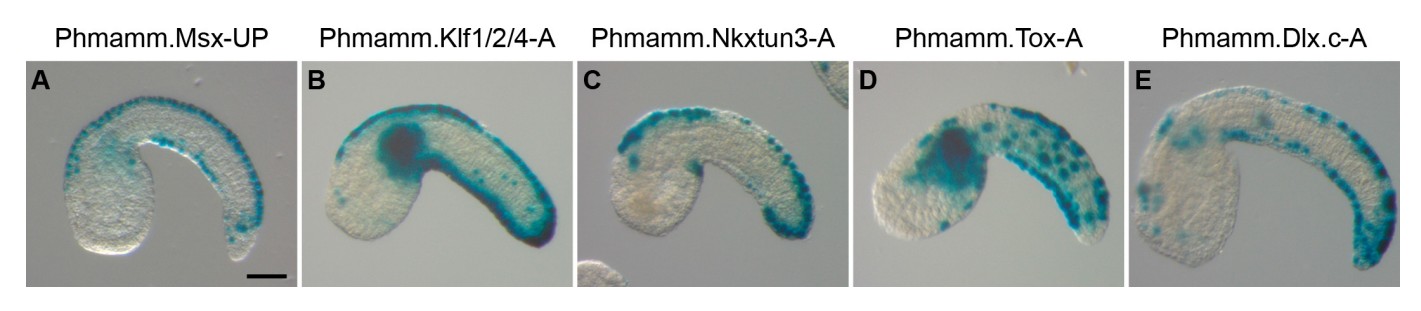

**Figure 5.** CRMs with activity in the tail epidermis midlines for *Phmamm.Msx*, *Phmamm.Klf1/2/4*, *Phmamm.Nkxtun3*, *Phmamm.Tox*, and *Phmamm.Dlx.c*. Representatives examples of X-gal staining at tailbud stages following *P. mammillata* embryos electroporation of *P. mammillata* genomic regions for *Msx* (A), *Klf1/2/4* (B), *Nkxtun3* (C), *Tox* (D), and *Dlx.c* (E). Embryos are shown in lateral view with dorsal to the top and anterior to the left. Scale bar: 50 μm.

The online version of this article includes the following figure supplement(s) for figure 5:

**Figure supplement 1.** CRMs controlling *Phmamm.Msx* expression in VDML.

**Figure supplement 2.** CRMs controlling *Phmamm.Klf1/2/4* and *Phmamm.Nkxtun3* expression in VDML.

**Figure supplement 3.** CRMs controlling *Phmamm.Tox* expression in VDML.

**Figure supplement 4.** CRMs controlling *Phmamm.Dlx.c* expression in VDML.

orthologous: KH2012:KH.C2.808 in *C. intestinalis* and Phmamm.g00005894 in *P. mammillata*) and the small intergenic region contains a proximal CRM active early in DML precursors and a distal CRM active late in VDML (*Figure 1* and *Figure 5—figure supplement 1*). TFBS analysis of the LED fragment suggested that *Phmamm.Msx* could be regulated like *Ciinte.Msx*: a direct target of Bmp signaling in the VML and of Msx in the DML (*Figure 5—figure supplement 1*).

### Klf1/2/4

We first isolated a 6.7 kb conserved region immediately upstream of *Klf1/2/4*, A, which was strongly active in VDML (*Figure 5B*, *Figure 5—figure supplement 2* and *Supplementary file 5*). When subdivided into three large pieces, two were active in VDML, suggesting a possible redundant regulation (*Figure 5—figure supplement 2*). The proximal region A3 (2.5 kb) was the region with strongest activity.

### Nkxtun3

We first isolated a 7.2 kb conserved region 2 kb upstream of *Nkxtun3*, A, which was strongly active in VDML (*Figure 5C*, *Figure 5—figure supplement 2* and *Supplementary file 5*). Similarly to *Phmamm-Klf1/2/4*, two of three sub-regions of A were active in VDML (*Figure 5—figure supplement 2*). In this case, a spatial difference of activity was observed: the distal region A1 (2.5 kb) was mainly active in the posterior half of VDML while the proximal region A3 (2.8 kb) was active throughout VDML. This latter region could be reduced to 1.5 kb (A3.1) with maintaining robust activity. However, a shorter fragment A3.3 (0.6 kb) was barely active.

### Tox

A short conserved region immediately upstream of *Tox*, A (1093 bp), was strongly active in VDML but also in tail muscle and mesenchyme (*Figure 5D*, *Figure 5—figure supplement 3* and *Supplementary file 5*). Activity in muscle/mesenchyme segregated to a proximal fragment, A2 (533 bp), while VDML activity segregated to a distal fragment, A1 (657 bp), which could be reduced to 240 bp (A1.1) with maintaining a robust VDML activity. The small size of these fragments allowed TFBS analysis (*Figure 5—figure supplement 3*) that suggested a possible regulation by Ascl.b as for *Ciinte.Tox* (*Figure 3—figure supplement 3*).

### Dlx.c

Based on RNA-seq profiling, we estimated that the predicted gene model for *Dlx.c* was lacking its first exon located approximately 8 kb upstream (*Supplementary file 5*). We thus isolated a 5.4 kb

region containing 2 blocks of conservation, A, which was strongly active in VDML (*Figure 5E* and *Figure 5—figure supplement 4*). While the distal region A1.2 (1.5 kb) containing one block of conservation was weakly active in VDML, the proximal one A3.1 (1205 bp) was robustly active in VDML as was a shorter version A3.2 (1034 bp) (*Figure 5—figure supplement 4*).

The results of this section indicate strong similarities between *C. intestinalis* and *P. mammillata* for caudal PNS formation based on gene expression and gene regulation.

## Conserved midline PNS gene regulation in Phlebobranchia

Here, we directly investigated whether the expression of caudal neurogenic midlines genes was regulated similarly between *C. intestinalis* and *P. mammillata*. To this end, we performed bi-directional 'enhancer swaps': we tested in *P. mammillata* embryos the activity of *C. intestinalis* VDML CRMs and vice versa. Strikingly, all the regions that we have tested were active in VDML in over 40–50% of the embryos in the recipient species (*Figure 6A–C* and *Figure 6—figure supplement 1*), the only exception being Phmamm.Nkxtun3-A3.1 that was already not strongly active in *P. mammillata* (*Figure 5—figure supplement 2*). Interestingly, conserved activity was not limited to VDML: for example, activity in the anterior palp region for Ciinte.Ascl.b-do1, Ciinte.Klf1/2/4-up1 and Ciinte.Dlx.c-A2 was observed when tested in both *C. intestinalis* and *P. mammillata* (*Figures 2B*, *3A and G*, 6Cii, 6Ciii, and 6Cvi).

We next extended such functional assays in two other species, *Phallusia fumigata* and *Ascidia mentula*, that like *C. intestinalis* and *P. mammillata* belong to the ascidian Phlebobranchia order (*Figure 6A*). A few CRMs (Ciinte.Nkxtun3-up7, Phmamm.Nkxtun3-A1 and Ciinte.Tox-int1) had weak activity, below 30%, in one or the other species. However, most tested regions were active in VDML in over 40–50% of the embryos independently of their origin (i.e. from *C. intestinalis* or from *P. mammillata*) (*Figure 6D–E* and *Figure 6—figure supplement 2*). As *C. intestinalis* is more divergent than *P. mammillata* when taking the recipient species as references, this suggests that gene regulation has remained unchanged despite almost 300 My of divergence time (*Figure 6A*).

We further investigated CRM activity conservation in Phlebobranchia by isolating the genomic region upstream of *Msx* for both *P. fumigata* and *A. mentula*. Both regions were active in VDML when electroporated in *C. intestinalis* or *P. mammillata* embryos (*Supplementary file 6*).

Testing the transcriptional activity of midline CRMs into four ascidian species strongly supports deep conservation of caudal midline gene expression regulation in Phlebobranchia.

## Minimal enhancers are not robust

Above results suggest a strong conservation of the caudal PNS GRN at the level of gene expression, gene regulation and its genomic encoding (CRMs) in Phlebobranchia ascidians. However, potentially homologous CRMs (CRMs from orthologous genes, located at similar positions in their locus and with conserved VDML activity in swap experiments), do not show significant DNA sequence conservation at large distance (i.e. between *C. intestinalis* and *P. mammillata*) (*Supplementary file 2* and *5*). There are previous examples in ascidians that clearly show that such conserved activity relies on conserved upstream regulators but that extensive binding site turnover explains the lack of sequence conservation (*Roure et al., 2014*; *Madgwick et al., 2019*; *Colgan et al., 2019*). We thus reasoned that the CRMs with conserved activity that we have isolated from *C. intestinalis* and *P. mammillata* could help in identifying conserved TFBS and shared TF regulating CRM activity. We also postulated that this approach would be effective with CRMs of small size. By applying a strict conservative approach on known putative regulators, we could hypothesize that *Msx* expression in the VDML from gastrula/neurula stages is directly regulated by Bmp signaling triggered by Admp in the VML, and by Msx itself in the DML (*Figure 1—figure supplement 2* and *Figure 5—figure supplement 1*). Similarly, *Tox* may be regulated directly by Ascl.b (*Figure 3—figure supplement 3* and *Figure 5—figure supplement 3*.).

We next focused on the proximal Ci-msxb-b6.5 line CRM that regulates early *Ciinte.Msx* expression in DML precursors (*Figure 1A*; *Roure et al., 2014*). We have previously shown that its size could be reduced from 707 bp to 273 bp (Ci-msxb-B) while maintaining a robust activity (around 50% of embryos with DML activity) (*Figure 7A*). Surprisingly, when the various deletions of Ci-msxb-b6.5 were tested in *P. mammillata*, they were barely active (<10% for constructs of 400 bp or less) (*Figure 7A*). Since these different constructs still contained the Otx and Smad-binding sites that are

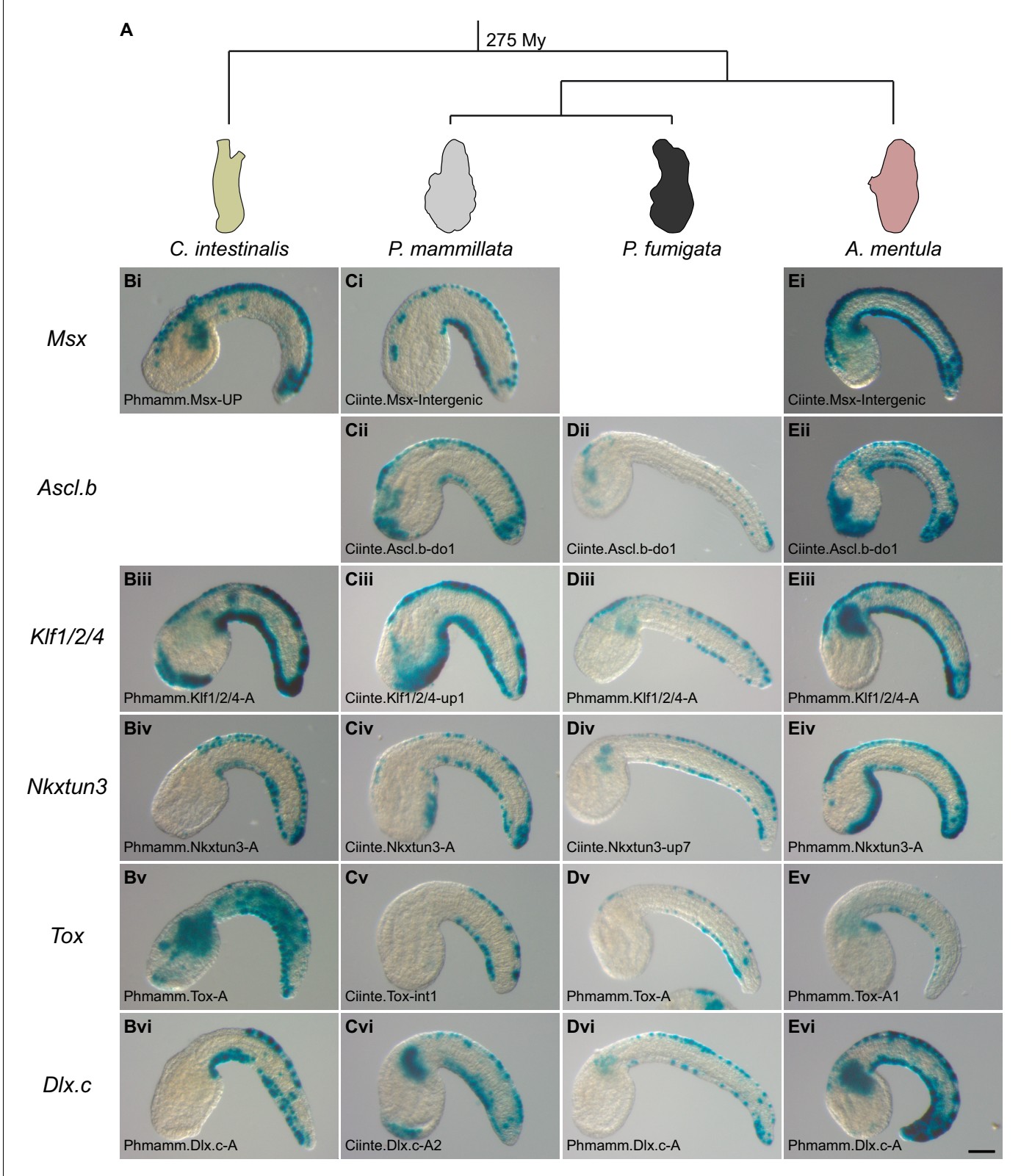

**Figure 6.** Conserved VDML activity during enhancer swaps in Phlebobranchia (**A**). Representative examples of X-gal stained embryos at tailbud stages following *C. intestinalis* (**B**), *P. mammillata* (**C**), *P. fumigata* (**D**), or *A. mentula* (**E**) embryo electroporation with *C. intestinalis* or *P. mammillata* CRMs. All shown CRMs for the genes *Msx* (i), *Ascl.b* (ii), *Klf1/2/4* (iii), *Nkxtun3* (iv), *Tox* (v), and *Dlx.c* (vi) are active in VDML in their species of origin. The name of

*Figure 6 continued on next page*

*Figure 6 continued*

the electroporated CRM is indicated on each picture. Details for each experiment can be found in *Figure 6—figure supplement 1* and *Figure 6—figure supplement 2*. Embryos are shown in lateral view with dorsal to the top and anterior to the left. Scale bar: 50 μm.

The online version of this article includes the following figure supplement(s) for figure 6:

**Figure supplement 1.** Conserved VDML activity during enhancer swaps between *C. intestinalis* and *P. mammillata*.

**Figure supplement 2.** *C. intestinalis* and *P. mammillata* midline CRMs are active in VDML of *P. fumigata* and *A. mentula* embryos.

essential for Ci-msxb-B activity in *C. intestinalis*, we wondered whether Ci-msxb-B was faithfully recapitulating *Ciinte.Msx* expression. To this end, we challenged Ci-msxb-B with known regulators of *Ciinte.Msx* expression (*Figure 7B*). Similarly to endogenous *Ciinte.Msx* expression, Ci-msxb-B was ectopically activated in posterior (b-line) ectoderm when the Fgf pathway was activated, and it was downregulated when the Nodal pathway was inhibited. However, upon Nodal pathway activation Ci-msxb-B was ectopically activated in posterior (b-line) ectoderm instead of anterior neurectoderm (a6.5 line); and Otx overexpression led to a repression of Ci-msxb-B instead of an activation.

We thus wondered whether other short CRMs that we have described here with VDML activity also behaved inappropriately when submitted to a genetic challenge. We thus determined the activity of these CRMs when a midline TF was overexpressed and compared the effects with what we have already described for the endogenous expression of the associated gene (*Roure and Darras, 2016*). We were very surprised to realize that while some interactions were identical, a large part was different (*Figure 7C*). Also, two different CRMs for *Ciinte.Nkxtun3* behaved differently.

We conclude from this section that short or minimal CRMs should be taken with cautious because they may not be robust to phylogenetic (enhancer swap) or genetic (overexpression) challenges.

## Variable levels of divergence in Stolidobranchia

Here we aimed at exploring whether the striking conservation of VDML GRN holds true outside Phlebobranchia. We first determined the expression patterns of several genes of this network during the development of two solitary Stolidobranchia species, *Molgula appendiculata* and *Halocynthia roretzi* (*Figure 8*, *Figure 8—figure supplements 1–3*). Both species are estimated to have diverged around 340 My ago, and 390 My ago with Phlebobranchia (*Figure 8A*; *Delsuc et al., 2018*). Using transcriptomic data for *M. appendiculata* and both transcriptomic and genomic data for *H. roretzi* (*Dardaillon et al., 2020*), we identified a single ortholog for each gene that we analyzed, except for *Ascl.b* for which two paralogs were found in *M. appendiculata*.

### Gene expression in *M. appendiculata*

For the orthologs of neurogenic midline genes, three had an expression very similar to their counterparts in *C. intestinalis* or *P. mammillata*. While *Moappe.Msx* and *Moappe.Klf1/2/4* were expressed throughout the VDML, *Moappe.Nkxtun3* expression was restricted to the posterior VDML (*Figure 8B, D and E* and *Figure 8—figure supplement 1*). *Moappe.Ascl.bβ* and *Moappe.Tox* were also expressed in the VDML but only in the posterior region and as spots (*Figure 8C and F* and *Figure 8—figure supplement 1*). By contrast, *Moappe.Ascl.bα* and *Moappe.Dlx.c* were not expressed in the VDML (*Figure 8G* and *Figure 8—figure supplement 1*). We also examined genes downstream of the neurogenic program, *Dlk* and *Celf3.a* (*Figure 8—figure supplement 1*). While *Moappe.Dlk* was expressed in mesoderm and central nervous system, we did not detect a specific expression in the VDML. *Moappe.Celf3.a* was expressed strongly in the central nervous system as in Phlebobranchia, and also in isolated spots on surface ectoderm that likely correspond to ectodermal sensory neurons. In particular, *Moappe.Celf3.a* was detected in two to five spots of the posterior VDML, suggesting that the number of CESNs in this species is small. Moreover, this caudal restriction of expression is in agreement with the expression of *Moappe.Nkxtun3*, *Moappe.Ascl.bβ*, and *Moappe.Tox*.

### Gene expression in *H. roretzi*

Among the orthologs of neurogenic genes, *Harore.Dlx.c* pattern had already been described as expressed in the VDML (*Wada et al., 1999*; *Figure 8M* and *Figure 8—figure supplement 2*). The other genes that we examined, *Harore.Msx*, *Harore.Ascl.b*, *Harore.Klf1/2/4*, *Harore.Nkxtun3*, and

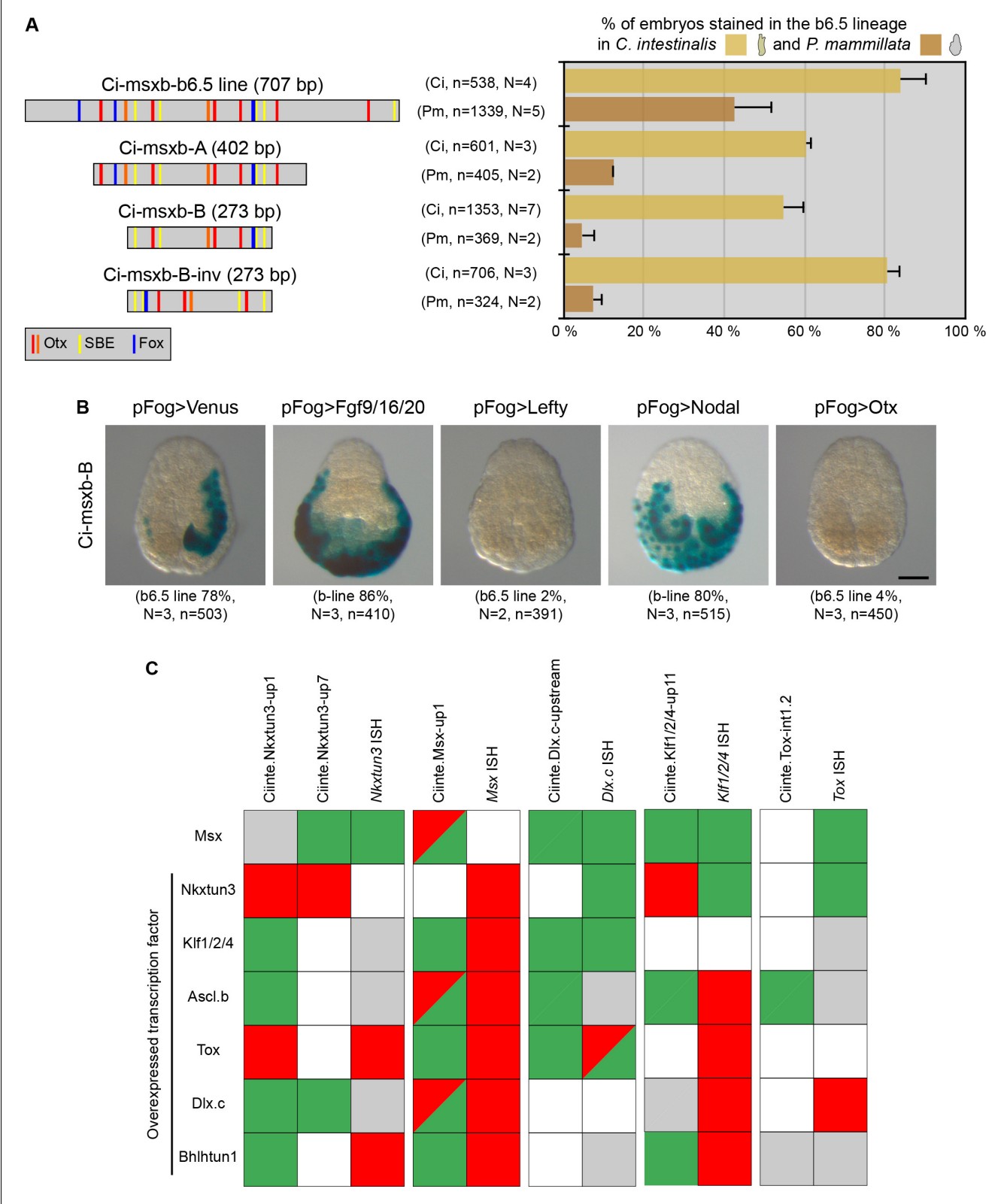

**Figure 7.** Minimal CRMs are not robust to phylogenetic and genetic challenges. (A) Structure of the early DML CRM 'Ci-msxb-b6.5 line' and its derivatives (adapted from *Roure et al., 2014*), and in vivo transcriptional activity in the embryos of *C. intestinalis* and *P. mammillata* (n indicates the total number of embryos examined, N indicates the number of independent experiments). (B) Effects of various overexpressions on Ci-msxb-B activity. The indicated factors (Fgf9/16/20, Lefty, Nodal and Otx) were expressed under the control of the early ectodermal driver pFog. Embryos at gastrula/
*Figure 7 continued on next page*

Figure 7 continued

neurula stages are shown in neural plate view with anterior to the top (n indicates the total number of embryos examined, N indicates the number of independent experiments). Scale bar: 50 µm. (C) Consequences of caudal midline TF overexpression on the activity of VDML CRMs. The effects were summarized as: activation (green), repression (red), no effect (gray), or not done (white). They were compared with results of endogenous gene expression from *Roure and Darras, 2016*.

*Harore.Tox*, were also expressed in the VDML (*Figure 8H–L* and *Figure 8—figure supplement 2*). Among them, *Harore.Ascl.b* and *Harore.Tox*, were not expressed throughout the VDML but rather in discrete regions or spots. CESNs have been described in *H. roretzi*, and they express *Celf3.a* (also known as *Etr*) (*Figure 8—figure supplement 2*; *Ohtsuka et al., 2001a*; *Ohtsuka et al., 2001b*; *Yagi and Makabe, 2001*). It is also known that their formation is regulated by the Notch pathway and the ligand *Delta* is expressed as spots in the VDML (*Akanuma et al., 2002*). We could also detect the expression of another Notch ligand *Dlk* in the VDML (*Figure 8—figure supplement 2*).

Overall, comparison of gene expression shows striking conservation in patterns but also in the temporal order despite almost 400 My of divergence between Stolidobranchia and Phlebobranchia. However, the degree of conservation is weaker when considering *M. appendiculata*.

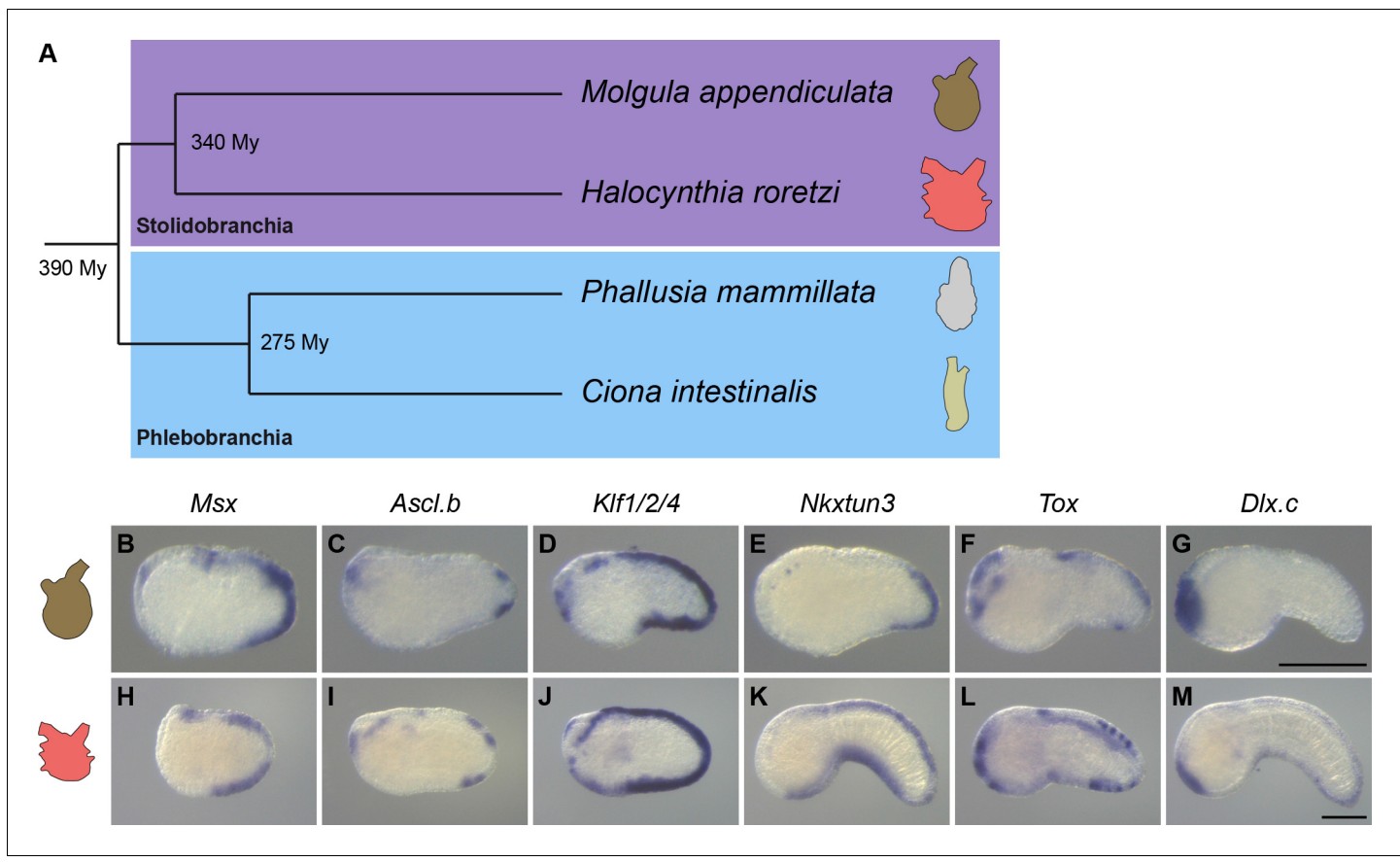

**Figure 8.** Overall conservation of midline gene expression in Stolidobranchia ascidians. (A) Phylogenetic relationships between *C. intestinalis*, *P. mammillata*, *M. appendiculata* and *H. roretzi* (estimated divergence times come from *Delsuc et al., 2018*). (B–M) In situ hybridization for *Msx* (B, H) at neurula stages, for *Ascl.b* (C, I), *Klf1/2/4* (D, J), *Nkxtun3* (E, K), *Tox* (F, L), and *Dlx.c* (G, M) at tailbud stages in embryos of *M. appendiculata* (B–G) and *H. roretzi* (H–M). Note that all genes except *Moappe.Dlx.c* are expressed in tail midlines. Embryos are shown in lateral view with dorsal to the top and anterior to the left. Scale bars: 100 µm.

The online version of this article includes the following figure supplement(s) for figure 8:

**Figure supplement 1.** Expression pattern of caudal PNS genes in *M. appendiculata*.
**Figure supplement 2.** Expression pattern of caudal PNS genes in *H. roretzi*.
**Figure supplement 3.** Comparison of midline TFs expression in *C. intestinalis*, *P. mammillata*, *M. appendiculata*, and *H. roretzi*.

## CRM swap

We next wondered whether conservation could extend to CRM activity. We thus tested various CRMs isolated from *C. intestinalis* and *P. mammillata* in *M. appendiculata* that is amenable to electroporation. A first qualitative analysis revealed that CRMs for several midline genes were found active in *M. appendiculata* VDML: *Msx* (*Figure 9A*), *Ascl.b* (*Figure 9B*), *Nkxtun3* (*Figure 9D*) and *Dlx.c* (*Figure 9F*). In other cases, CRMs were active but not in VDML: anterior ectoderm and central nervous system for Ciinte.Klf1/2/4-up1 (*Figure 9C*), or tail muscle for Phmamm.Tox-A (*Figure 9E*). However, when the strength of the activity was considered (measured indirectly as the percentage of stained embryo in VDML), the situation appeared quite different. VDML activity was seldom observed for CRMs from *Msx* and *Ascl.b* (*Figure 9—figure supplement 1*). These were only large CRMs for *Nkxtun3* and *Dlx.c* that were robustly active in *M. appendiculata* VDML. The overall low levels of activity for the tested CRMs could be explained by a non-optimal electroporation procedure. While this cannot completely be ruled out, electroporation of the regulatory sequences for the ubiquitous gene *Cirobu.Ef1α* (*Feinberg et al., 2019*) led to strong expression of the *LacZ* reporter gene (89% of stained embryos, n = 74 in two independent experiments) and suggested a rather efficient procedure. Interestingly, for both *Nkxtun3* and *Dlx.c*, CRMs from both *C. intestinalis* and *P. mammillata* were active in *Molgula* VDML, supporting the specificity of the results. This further suggests that within the 6-genes network that we have explored, only these two genes have retained conserved regulation.

To further explore whether the global lack of conservation that we have observed in *M. appendiculata* could be a generality in Stolidobranchia, we have isolated genomic upstream regions for *Msx*, *Nkxtun3*, and *Tox* from *H. roretzi* (*Figure 9—figure supplement 2* and *Supplementary file 7*). The three regions were poorly active in *M. appendiculata* embryos except Harore.Nkxtun3-B that was active in anterior sensory vesicle in up to 29% of the embryos (*Figure 9I*). Only Harore.Msx-A had a striking VDML pattern but in a minority of embryos (*Figure 9G*). These results could simply be that these regions do not behave as VDML CRMs since we have not determined their activity in *H. roretzi*. However, in *P. mammillata*, while Harore.Nkxtun3-B was weakly active in VDML with additional staining in lateral tail epidermis (*Figure 9J* and *Figure 9—figure supplement 2*), both Harore.Msx-A and Harore.Tox-A were specifically and strongly active in VDML (*Figure 9H and L* and *Figure 9—figure supplement 2*).

## Discussion

### Regulation of midline gene expression in *C. intestinalis*

We have reported the identification of midline CRMs for six out of the seven genes that we have tackled. Our approach was simply based on isolating conserved regions (with *C. savignyi*) close to the gene of interest. Similarly to numerous reports, active CRMs were found mostly within the first few kilobases immediately upstream of the gene (a single region was located downstream). Interestingly, the largest CRMs that we have tested contain activity for several expression domains where the gene is expressed and do not usually display ectopic activity. We further reduced the size of these CRMs through classical 'promoter bashing' guided by sequence conservation. Activity in different domains usually partitioned during this deletion process while we have not documented this in detail (for example for *Dlx.c*, palp expression in the large region Ciinte.Dlx.c-A2 is lost from the small region Ciinte.Dlx.c-upstream; *Figure 3G,H*). Interestingly, the conserved active CRMs that we have identified systematically matched with open chromatin as determined by enrichment in ATAC-seq (*Figures 1* and *2*, *Figure 3—figure supplements 1–4*, and *Supplementary file 2*; *Madgwick et al., 2019*). Tissue- and stage-specific ATAC-seq data should certainly facilitate active CRM discovery and allow exhaustive identification of CRMs regulating a given expression pattern.

Although our approach was not exhaustive, important conclusions could be drawn. First, the seemingly continuous expression of *Msx* in the b6.5 lineage and its derivative, the DML, is actually under the control of two separate CRM: an early proximal element active from the 64-cell stage (*Roure et al., 2014*) and a late distal element active from gastrula stages (*Figure 1*). TFBS analysis using our current understanding of *Msx* regulation (*Figure 1—figure supplement 1*) supports a model of maintenance of gene expression by autoregulation whereby Msx protein produced through the early proximal CRM activates late *Msx* expression via the distal CRM (*Figure 1—figure*

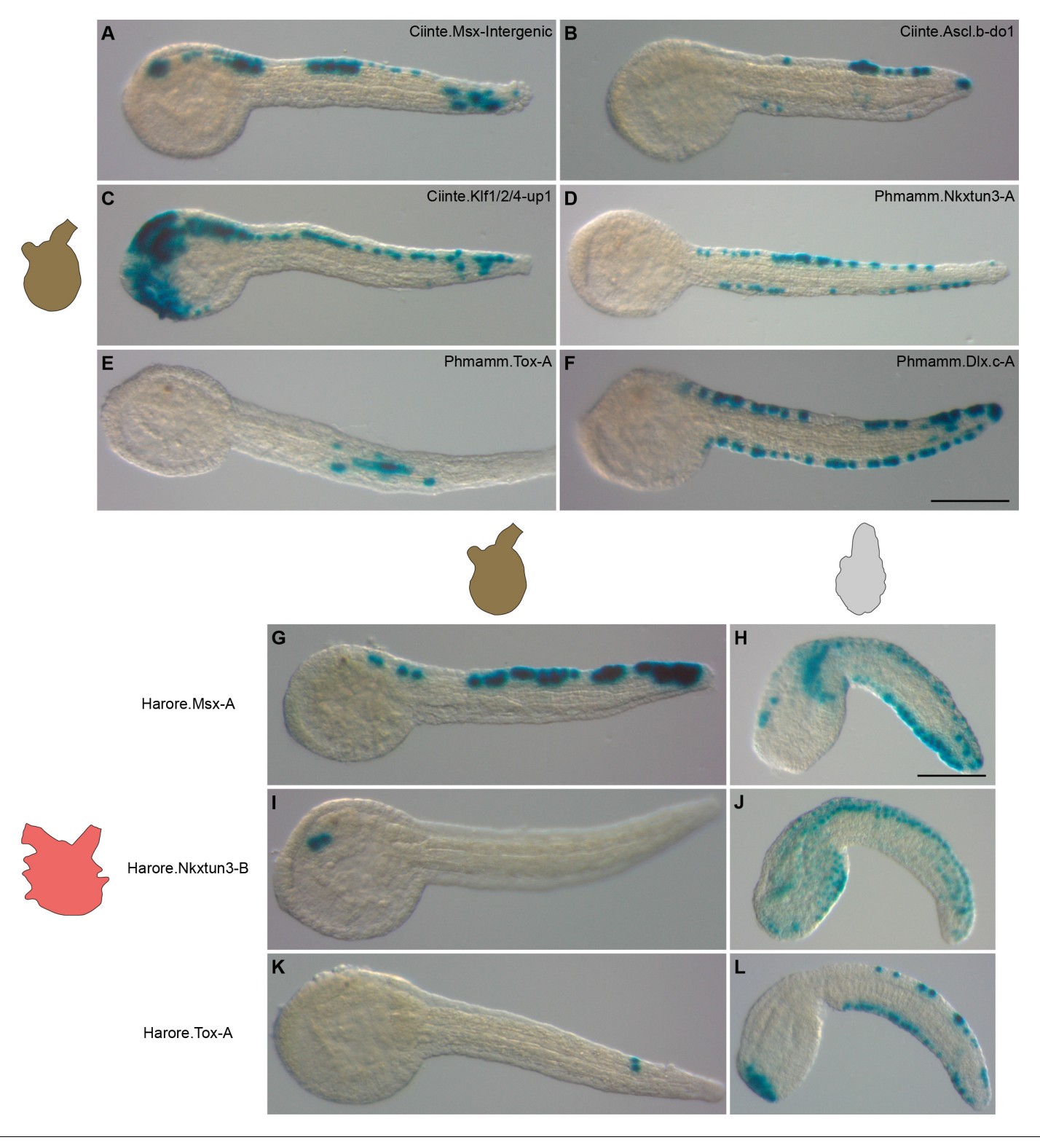

**Figure 9.** Divergence in gene regulation in Stolidobranchia ascidians. (A–F) Activity of various CRMs from *C. intestinalis* and *P. mammillata* in the embryos of *M. appendiculata*. The name of the electroporated CRM is indicated on each picture. Details for each experiment can be found in *Figure 9—figure supplement 1*. (G–L) Activity of various genomic regions from *H. roretzi* in the embryos of *M. appendiculata* (G, I and K) and *P. mammillata* (H, J and L). Details for each experiment can be found in *Figure 9—figure supplement 2*. Embryos are shown in lateral view with dorsal to the top and anterior to the left. Scale bars: 100 μm.

*Figure 9 continued on next page*

eLife Research article

Developmental Biology | Evolutionary Biology

**Figure supplement 1.** Activity of VDML CRMs from *C. intestinalis* and *P. mammillata* in the embryos of *M. appendiculata*.

**Figure supplement 2.** Activity of genomic regions from *H. roretzi* in the embryos of *P. mammillata* and *M. appendiculata*.

*supplement 2*). Second, we could classify the genes in two classes: *Msx* and *Ascl.b* with uncoupling of regulation in dorsal and ventral midlines (*Figures 1* and *2*); and *Klf1/2/4*, *Nkxtun3*, *Tox* and *Dlx.c* regulated in a 'pan-midline' manner (*Figure 3—figure supplements 1–4*). These observations are in agreement with previous predictions based on regulatory interactions (*Pasini et al., 2006*; *Roure and Darras, 2016*; *Feinberg et al., 2019*; *Waki et al., 2015*): *Msx* and *Ascl.b* are upstream regulators that integrate inputs from the dorsal and ventral midline inducers and regulate the expression of the other four downstream factors. Surprisingly, we did not identify VML-specific CRMs. While the reason is still unclear, a possible explanation would be that there is a physical overlap between VML and DML CRMs with the DML being embedded in the VML CRM. Prediction of TFBS for putative *trans* acting factors in the *Msx* VDML minimal CRM (Ciinte.Msx-up10) supports this hypothesis: binding sites for dorsal and ventral factors are intermingled or overlapping (*Figure 1—figure supplement 2*). Targeted mutagenesis of these putative binding sites should enable testing such an hypothesis. Third, analysis of conserved TFBS in small active CRMs provides interesting hypotheses that will need to be tested experimentally using finer and possibly quantitative approaches. Although both *Msx* and *Ascl.b* are likely to integrate initial inducing cues, they do not seem to be regulated similarly: *Msx* is possibly a direct Bmp signaling target in the VML like vertebrate *Msx* in some tissues (*Esteves et al., 2014*; *Brugger et al., 2004*), but *Ascl.b* is not (*Figure 1—figure supplement 2* and *Figure 2—figure supplement 1*). *Klf1/2/4* and *Nkxtun3* are possibly directly activated by Msx, and *Tox* by Ascl.b. Finally, it is most likely that additional TFs remain to be identified to understand gene regulation in the VDML since the activity of several CRMs and deletions cannot be explained using the TFs that we currently know.

## High conservation of PNS formation in Phlebobranchia

Expression patterns, overexpression experiments and CRMs identification in *P. mammillata* showed high similarities with *C. intestinalis* (*Figures 4*, *5* and *10*). This was further demonstrated by CRM swap experiments (*Figures 6* and *10*) where midline CRMs from *C. intestinalis* were active in *P. mammillata* midlines and vice versa, and also in two additional Phlebobranchia species (*Figure 6*).

These observations suggest that orthologous genes are regulated similarly and that CRMs are controlled using the same regulatory logic. Since non-coding DNA does not show sequence conservation, for example between *C. intestinalis* and *P. mammillata* CRMs that we identified (*Supplementary file 2* and *5*), it is difficult to define homologous CRMs. This is likely explained by extensive turnover of the binding sites for upstream transcription factors, a situation that we have previously described for the regulation of the early *Msx* CRMs by Otx and Nodal (*Roure et al., 2014*). The newly identified *Msx* CRM from *A. mentula* actually contains a region dense in Otx and Smad binding sites like their counterparts in *Ciona* and *Phallusia* (not shown). This phenomenon has been described for other CRMs between *C. intestinalis* and other species and outside ascidians (*Buffry et al., 2016*; *Colgan et al., 2019*; *Oda-Ishii et al., 2005*).

Given the modular nature of *cis*-regulation and the increased recognition of the existence of 'redundant', 'shadow' or distributed enhancers (*Barolo, 2012*; *Cannavò et al., 2016*), namely multiple CRMs with seemingly similar activity for a given gene, caution has to be taken for pairwise comparisons in TFBS composition of genomic pieces originating from different species. We could nevertheless apply the above approach for two other CRMs. In both *Ciona* and *Phallusia*, the late *Msx* CRMs with VDML activity are distal and share binding sites for Msx and Bmp regulation (SBE and BRE) (*Figure 1—figure supplement 2* and *Figure 5—figure supplement 1*). CRMs for *Tox* are localized upstream, close to the transcription start in both species (at least for one of the two isoforms in *C. intestinalis*), and are characterized by an enrichment in E-box sites, possibly mediating activation by Ascl.b (*Figure 3—figure supplement 3* and *Figure 5—figure supplement 3*). These observations are in agreement with conserved regulation at the base of Phlebobranchia and we propose that the above CRMs are orthologous.

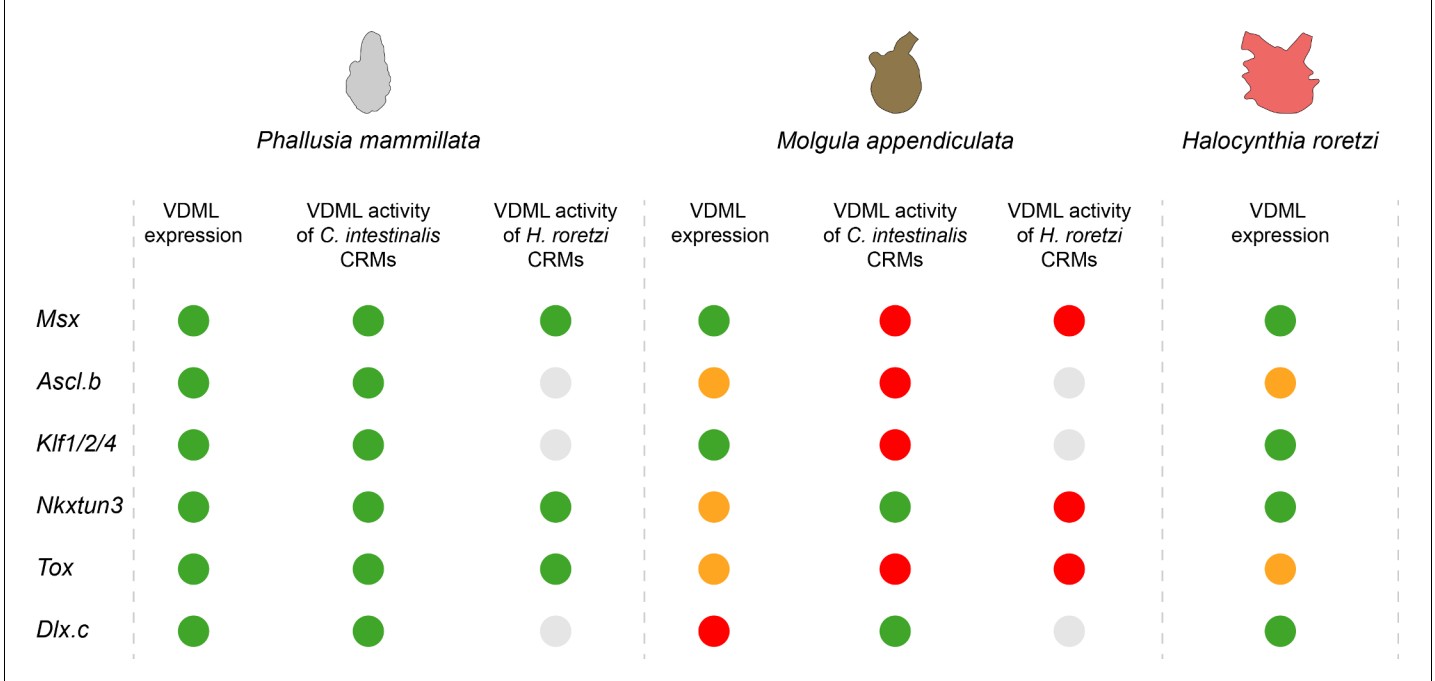

**Figure 10.** Summary of comparative results obtained in the present study. For each gene, results of in situ hybridization and enhancer swaps are summarized. VDML expression (in situ hybridization) or activity (transcriptional assay) is represented as a green circle, expression in part of the VDML as an orange circle, and lack of expression or activity as a red circle. Gray circle: not done.

We thus aimed at using such a strategy using candidate orthologous minimal midline CRMs to define shared motifs and identify *trans*-acting factors. However, the 'swap' and overexpression experiments described in *Figure 7* refrained us from pursuing since such CRMs were not robust to phylogenetic and genetic perturbations. Whether these observations are general to ascidian minimal CRMs remains to be explored. A possible interpretation could be linked to the modular nature of *cis*-regulation. The largest CRMs that we tested are likely to contain most, if not all, the regulatory information for VDML expression. By contrast, shorter elements contain only a subset of the regulatory information, and would consequently not behave identically to the gene following genetic challenge. The fact, that such short elements may not be active when placed in another species while larger regions are, suggests that drift has occurred at the level of individual modules making them silent. Independently of our current understanding of this phenomenom, reciprocal in vivo transcriptional assays between *Ciona* and *Phallusia* offer a direct way to identify and determine the properties of the DNA elements that confer robustness.

## Global conservation of PNS formation in ascidians?

In situ hybridization in two members of the Stolidobranchia revealed strong conservation in gene expression. However, while the patterns were virtually identical between *Ciona* and *Phallusia*, we could detect several differences (*Figures 8* and *10*). First, in both *Molgula* and *Halocynthia*, the genes *Ascl.b* and *Tox* were not expressed throughout the VDML but had a spotted expression. Such an expression pattern is correlated with the number and location of future PNS neurons. As shown in *Figure 8—figure supplements 1–2* and already known for *Halocynthia*, presumptive neurons labeled by *Celf3.a* expression are present in a very small number and only in the posterior tail, at least on the ventral side, when compared to *Ciona* (*Pasini et al., 2006*; *Ohtsuka et al., 2001b*; *Yagi and Makabe, 2001*; *Akanuma et al., 2002*). This would suggest a variation at the level of the Stolidobranchia family and that the function of *Ascl.b* and *Tox* would possibly be different from the other 'pan-midline' transcription factors and from their orthologs in Phlebrobranchia. Interestingly, in Phlebrobranchia, we have seen above a possible direct regulation of *Tox* by Ascl.b, suggesting that

PNS formation in ascidians could be controlled by the following VDML sub-networks: *Ascl.b* and *Tox* on one side, and *Msx*, *Klf1/2/4*, *Nkxtun3*, and *Dlx.c* on the other side. Second, in *Molgula*, *Nkxtun3* expression was restricted to the posterior VDML. Third, *Moappe.Dlx.c* was not expressed in the VDML, while palp expression was conserved (*Figure 8—figure supplement 3*). These two latter observations suggest a genus- or species-specific variation.

When focusing on gene regulation, we detected again stronger conservation within Phlebobranchia than between Phlebobranchia and Stolidobranchia (*Figure 10*). This is particularly true when considering large CRMs of several kilobases that have better chance of capturing the widest regulatory information. This difference between ascidian orders could be the result of increased drift because of larger evolutionary distance and/or because of discrete changes linked to taxonomical level change. We were nevertheless surprised to see that CRMs from *Halocynthia* were active in midlines when tested in *Phallusia* but not in *Molgula* (*Figures 9* and *10*). Although we have obtained these results on a limited number of CRMs and while we have not tested CRMs from *M. appendiculata* whose genome has not been sequenced, we believe that, taking expression patterns into account, *M. appendiculata* is the divergent species among all species that we have used (*Figure 10*). Our hypothesis is in agreement with previous CRM swap experiments between *Ciona* and other *Molgula* species (*Stolfi et al., 2014*), with the facts that *Molgula* is a fast evolving genus within ascidians (*Delsuc et al., 2018*; *Alié et al., 2018*) and that Molgulids is a family where loss of chordate body plan has occurred several times independently (tail-less larvae) (*Huber et al., 2000*).

Surprisingly, we did not detect obvious correlations between points of conservation in gene regulation and gene expression or GRN structure. *Nkxtun3* and *Dlx.c* are the two genes whose CRMs are active in *Molgula* midlines (*Figure 9* and *Figure 9—figure supplement 1*). They are not the genes with the best conserved expression; and while *Dlx.c* is in a downstream position of the network, *Nkxtun3* has an intermediate position (*Roure and Darras, 2016*). Interestingly, in *M. appendiculata* *Nkxtun3* is found only in posterior VDML and *Dlx.c* is not expressed in the VDML, but CRMs from both *Ciona* and *Phallusia* were active throughout *M. appendiculata* VDML. Consequently, it is likely that upstream factors are likely expressed throughout the VDML and that changes in the *cis*-regulatory landscapes have occurred for both genes in *M. appendiculata*. Similarly, a CRM from *Halocynthia Tox* (expressed in a spotted manner) was active throughout the VDML in *Phallusia*.

Given the phylogenetic positions of the species (*Figure 8A*) and the divergent behavior of *Molgula* (see above), a parsimonious analysis of our results (summarized in *Figure 10*) implies that PNS gene expression and regulation are globally conserved in ascidians despite extensive genomic divergence in the course of almost 400 My of evolution. This is a similar time window that separates mammals from teleosts (*Delsuc et al., 2018*), and several developmental processes including CRM activity have been shown conserved between mouse and zebrafish for example, or even between more divergent vertebrate species (zebrafish and lamprey) (*Hockman et al., 2019*; *Fisher et al., 2006*; *Gehrke et al., 2015*; *McGaughey et al., 2008*). While some of the cited examples refer to CRM conserved activity despite lack of sequence conservation, in vertebrates CRMs are usually associated with sequence conservation. In ascidians, the situation is the opposite with an absence of non-coding DNA conservation and lack of synteny.

We would like to conclude by stressing two operational points for comparative studies in ascidians that may be relevant more broadly. The CRMs are interesting tools to probe gene regulation especially during swap experiments in species with efficient transgenesis. However, the lack of fidelity of small CRMs that we have uncovered may blur comparative conclusions. Finally, the choice of species for comparisons is also essential. While divergent species are highly informative to understand evolution at work, they may lead to inaccurate conclusions at a broader level. Although the established model species *C. intestinalis* and *H. roretzi* appear to be relevant species to instruct the status of ancestral ascidians, we would advocate for broad taxonomic sampling in functional comparative studies.

## Materials and methods

### Embryo obtention and manipulation

Adults from *C. intestinalis* (formerly referred to *Ciona intestinalis* type B *Brunetti et al., 2015*) and *Ascidia mentula* were provided by the Centre de Ressources Biologiques Marines in Roscoff

(EMBRC-France). *Phallusia fumigata* and *Molgula appendiculata* were provided by the Centre de Ressources Biologiques Marines in Banyuls-sur-mer (EMBRC-France). *Phallusia mammillata* were collected during trawling by professional fishermen in the Banyuls-sur-mer area. Gametes were collected from the gonoducts, except for *M. appendiculata* where gonads dissociation released sperm and immature eggs that spontaneously matured in sea water within 20–30 min. Egg dechorionation was performed before fertilization for *P. fumigata* and *A. mentula* as previously described for *P. mammillata* (*Roure et al., 2014*), and after fertilization for *M. appendiculata* as described for *C. intestinalis* (*Mita-Miyazawa et al., 1985*). Plasmid DNA (50 µg) was electroporated as described previously (*Feinberg et al., 2019*) through a single 32 ms electrical pulse with the following voltage: 20V for *M. appendiculata*; 25V for *C. intestinalis*; and 37V for *P. mammillata*, *P. fumigata*, and *A. mentula*.

P. *mammillata* embryos were treated with 150 ng/ml recombinant zebrafish Bmp4 protein (1128-BM, R&D Systems Inc, stock solution in HCl 4 mM + BSA 0,1%) complemented with 0.1% BSA from the eight-cell stage, with 20 µM of the Bmp receptor inhibitor Dorsomorphin (S7306, Euromedex, stock solution in water) from early gastrula stages or with 25 µM of the γ-secretase inhibitor DAPT (D5942, Sigma-Aldrich, stock solution in DMSO) from early neurula stages. Control embryos were grown in seawater containing 0.1% BSA and 0.5% DMSO. Whole mount in situ hybridization were performed as described (*Feinberg et al., 2019*) with Dig-labeled probes synthesized from clones described in *Supplementary file 8*. These clones were selected from EST libraries or PCR-amplified based on genomic or/and transcriptomic data (*Supplementary file 9*; *Dardaillon et al., 2020*; *Kawashima et al., 2000*; *Kawashima et al., 2002*). Reference transcriptome for *M. appendiculata* was generated from RNA-seq data on mixed embryonic stages and will be described elsewhere. Ten to 15 control embros per stage were examined for normal expression patterns, whereas 40–60 embryos were analyzed in the case of perturbations.

## In vivo transcriptional assay

Genomic regions were PCR amplified and placed upstream of the minimal promoter of the *Ciinte. Fog* gene (KH2012:KH.C10.574) (except for three regions directly abutting the transcribed region of the gene: Ciinte.Nkxtun3-upstream, Ciinte.Ascl.b-upstream, and Ciinte.Bhlhtun1-upstream) and the *LacZ* reporter gene using the Gateway technology (Invitrogen) as described previously (*Roure et al., 2007*). Details for newly isolated genomic regions, primers, templates, and vectors are available in *Supplementary file 1*. Details for already described regions (*Figure 1* and *Figure 5—figure supplement 1*) can be found in *Roure et al., 2014*. In the case of *Msx* in *A. mentula*, for which no whole genome sequence is available, we reasoned that local synteny would be conserved given the phylogenetic relationships between *Ascidia*, *Phallusia*, and *Ciona* genera (*Figure 6A*). Since the gene upstream of *Msx* is orthologous in sequenced *Ciona* and *Phallusia* species (KH2012:KH.C2.808 in *C. intestinalis*; Phmamm.g00005894 in *P. mammillata*), we amplified the *Msx* upstream intergenic region using degenerate primers defined in the coding sequences of *Msx* and its upstream neighbor (Fwd: TTYGAYAARTAYCARTTYGA; Rev: TTYTCYTGRAAYTTRTTYTC). Sequencing of this amplicon led to designing new primers for PCR amplification and cloning (*Supplementary file 1*). The sequence of the amplified region, Asment.Msx-up, can be found in *Supplementary file 9*.

Following electroporation and development, embryos were fixed at the desired stage and stained for β-galactosidase activity using X-gal (*Roure et al., 2014*). For each experiment, the number of embryos with β-galactosidase-positive cells in VDML was scored (a total of 30–300 embryos were counted per experiment). The number of experiments performed and the number of embryos examined is described in the text and figures. Graphs in the figures represent average values, with error bars denoting the standard deviation.

In the case of concomitant overexpression of a TF in *C. intestinalis* (*Figure 7*), at least two independent experiments have been performed, but without scoring. Constructs for TF overexpression have been previously described (*Roure and Darras, 2016*).

## Gene model identifiers

The genes described in this study were named according to the nomenclature in the tunicate community (*Stolfi et al., 2015*) and this may differ from previous publications. Homologous genes were identified using blast against genome/transcriptome and pre-computed orthologies when available

from the Aniseed database (*Dardaillon et al., 2020*). Gene identifiers come from the following assemblies: *C. robusta* KH2012, *P. mammillata* MTP2014, *P. fumigata* MTP2014, and *H. roretzi* MTP2014 (*Supplementary file 1*). In some cases, gene or transcript models were absent or incorrect; we tentatively built from RNA-seq and ESTs transcript models that are available in *Supplementary file 9*.

### Identification of putative TFBS

We first generated a tentative GRN for midline gene transcriptional regulation (*Figure 1—figure supplement 1A*) by incorporating known gene function and interactions (*Roure et al., 2014*; *Pasini et al., 2006*; *Roure and Darras, 2016*; *Waki et al., 2015*; *Joyce Tang et al., 2013*; *Bertrand et al., 2003*) and by formulating the following hypotheses: dorsal expression of the earliest genes, *Msx* and *Ascl.b*, may be regulated by the genes expressed earlier in DML precursors, namely *Msx* itself and *Dlk* (Notch ligand); ventral expression of the earliest genes, *Msx* and *Ascl.b*, may be regulated directly by Bmp signaling or by Bmp-activated genes/VML genes, namely *Tbx2/3*, *Nkxtun1*, *Nkx2-3/5/6* and *Irx.c* (*Waki et al., 2015*; *Imai et al., 2004*); and midline expression of the other genes may be regulated by all midline factors expressed before. We also considered autoregulation. For each factor or pathway (except *Tox* that is thought to bind DNA in a sequence-independent manner [*O'Flaherty and Kaye, 2003*]), we assigned consensus or specificity matrices obtained for the *C. intestinalis* factor or orthologous gene(s) from human, mouse, or fly (identified from the Aniseed database). Matrices were retrieved from Aniseed, Jaspar, and CIS-BP databases (*Dardaillon et al., 2020*; *Nitta et al., 2019*; *Weirauch et al., 2014*; *Fornes et al., 2020*; *Yao et al., 2006*) and are displayed in *Figure 1—figure supplement 1B*. For CRM shorter than 1 kb, sequences and their counterparts from the sister species (*C. robusta* and *C. savignyi*; *P. mammillata* and *P. fumigata*) were retrieved from Aniseed and scanned using FIMO (http://meme-suite.org/tools/fimo) with a match p-value threshold of 0.001 (*Grant et al., 2011*). TFBS were considered conserved when they were found close to each other (within approximately 60 bp) in the aligned sequences (alignment performed using zPicture: https://zpicture.dcode.org/; *Ovcharenko et al., 2004*). Only conserved sites were further considered in the analysis.

## Acknowledgements

We thank EMBRC-France (Banyuls-sur-mer and Roscoff marine stations) and G Diaz (Port-Vendres) for providing animals. We are grateful to C Labrune, JM Amouroux and F Monniot for help in identifying *M. appendiculata*. Many thanks to the *Phallusia* genome consortium (H Yasuo, A McDougall and P Lemaire) for access to various resources (genomes, transcriptomes and cDNA clones).

## Additional information

### Funding

| Funder | Grant reference number | Author |
|---|---|---|
| Agence Nationale de la Recherche | ANR-11-JSV2-007 | Sébastien Darras |
| Agence Nationale de la Recherche | ANR-17-CE13-0027 | Sébastien Darras |
| Fondation des Treilles | | Joshua F Coulcher |
| Centre National de la Recherche Scientifique | DBM2020 | Sébastien Darras |

The funders had no role in study design, data collection and interpretation, or the decision to submit the work for publication.

### Author contributions

Joshua F Coulcher, Conceptualization, Data curation, Validation, Investigation, Methodology; Agnès Roure, Conceptualization, Resources, Data curation, Validation, Investigation, Methodology; Rafath

Chowdhury, Conceptualization, Investigation, Methodology; Méryl Robert, Laury Lescat, Conceptualization, Investigation; Aurélie Bouin, Juliana Carvajal Cadavid, Investigation; Hiroki Nishida, Resources; Sébastien Darras, Conceptualization, Data curation, Supervision, Funding acquisition, Investigation, Methodology, Writing - original draft, Project administration

**Author ORCIDs**
Hiroki Nishida [ORCID] http://orcid.org/0000-0002-7249-1668
Sébastien Darras [ORCID] https://orcid.org/0000-0002-0590-0062

**Decision letter and Author response**
Decision letter https://doi.org/10.7554/eLife.59157.sa1
Author response https://doi.org/10.7554/eLife.59157.sa2

---

## Additional files

**Supplementary files**

• Supplementary file 1. List of all genomic regions tested in the present study.

• Supplementary file 2. Genome browser view for each locus of the seven caudal PNS neurogenic TFs in *Ciona robusta*. Tested CRMs were added to the data extracted from the Aniseed website (https://www.aniseed.cnrs.fr/; *Dardaillon et al., 2020*).

• Supplementary file 3. Activity of various genomic regions for the gene *Ciinte.Bhlhtun1*. (Top panel) Snapshot of the *Ciinte.Bhlhtun1* locus depicting ATAC-seq profile at mid-neurula stages, tested genomic regions, transcript models and conservation between *C. robusta* and *C. savignyi* (from https://www.aniseed.cnrs.fr/ and *Dardaillon et al., 2020*; *Madgwick et al., 2019*). (Middle panel) Representative examples of X-gal-stained embryos at tailbud stages following electroporation of Ciinte.Bhlhtun1-upstream, Ciinte.Bhlhtun1-up1 and Ciinte.Bhlhtun1-down1. Embryos are shown in lateral view with dorsal to the top and anterior to the left. Scale bar: 100 µm. (Bottom panel) Schematic representation of the various constructs and their activity at tailbud stages in DML (blue) and VML (purple) (n indicates the total number of embryos examined, N indicates the number of independent experiments). Note that while VDML activity is rare, activity can be detected at other sites of endogenous *Ciinte.Bhlhtun1* expression: anterior epidermis around the palps for Ciinte.Bhlhtun1-up1, and notochord and stomodeum for Ciinte.Bhlhtun1-down1.

• Supplementary file 4. Activity of various genomic regions for the genes *Phmamm.Ascl.b* and *Phmamm.Bhlhtun1*. Snapshots of the *Phmamm.Ascl.b* and *Phmamm.Bhlhtun1* loci depicting ATAC-seq profile at mid-neurula stages, tested genomic regions, transcript models and conservation between *P. mammillata* and *P. fumigata* (from https://www.aniseed.cnrs.fr/ and *Dardaillon et al., 2020*; *Madgwick et al., 2019*). Representative examples of X-gal stained embryos at tailbud stages following electroporation of Phmamm.Ascl.b-A (no activity), Phmamm.Ascl.b-B (activity in palps and anterior nervous system), and Phmamm.Bhlhtun1-A (activity in notochord, endodermal strand and tail tip) into *P. mammillata* embryos. Embryos are shown in lateral view with dorsal to the top and anterior to the left. Scale bar: 50 µm. Schematic representation of the various constructs and their activity at tailbud stages in VDML (blue) (n indicates the total number of embryos examined, N indicates the number of independent experiments). Note that while VDML activity is rather robust for Phmamm.Bhlhtun1-A, it was not considered further since this activity was restricted to the very posterior cells of the midlines at the tail tip.

• Supplementary file 5. Genome browser view for each locus of the seven caudal PNS neurogenic TFs in *Phallusia mammillata*. Tested CRMs and predicted cDNAs were added to the data extracted from the Aniseed website (https://www.aniseed.cnrs.fr/; *Dardaillon et al., 2020*).

• Supplementary file 6. Identification of VDML CRMs for *Phfumi.Msx* and *Asment.Msx* genes. (Top) Snapshot of the *Phfumi.Msx* locus. (Middle) Activity of *Phfumi.Msx* and *Asment.Msx* CRMs at tailbud stages in VDML (blue) of *C. intestinalis* and *P. mammillata* embryos (n indicates the total number of embryos examined, N indicates the number of independent experiments). (Bottom) Representative

examples of X-gal staining at tailbud stages (embryos in lateral view with dorsal to the top and anterior to the left, scale bar: 50 µm).

• Supplementary file 7. Genome browser view for each locus for three caudal PNS neurogenic TFs in *Halocynthia roretzi*. Tested CRMs were added to the data extracted from the Aniseed website (https://www.aniseed.cnrs.fr/; *Dardaillon et al., 2020*).

• Supplementary file 8. List of DNA clones used for in situ hybridization.

• Supplementary file 9. Various sequences. Predicted cDNA sequences for genes in *C. intestinalis, P. mammillata* and *M. appendiculata*: transcripts models from RNA-seq data and ESTs sequences were used to build the cDNA sequences. Open reading frame is highlighted in bold. Sequence of the genomic region Asment.Msx-up isolated from *A. mentula*.

• Transparent reporting form

### Data availability

All data generated or analyzed during this study are included in the manuscript and supporting files.

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
