## [Decision Letter]

**Acceptance summary:**

The authors of this study focused on four ascidian species with more than 250 million years of evolutionary divergence and performed an impressive suite of laboratory experiments and DNA sequence analyses on key molecular factors forming the peripheral nervous system of the ascidians. They report possible conservation of the regulatory mechanism over this evolutionary period for the first time.

**Decision letter after peer review:**

Thank you for submitting your article "Conservation of peripheral nervous system formation mechanisms in divergent ascidian embryos" for consideration by *eLife*. Your article has been reviewed by three peer reviewers, one of whom is a member of our Board of Reviewing Editors, and the evaluation has been overseen by Marianne Bronner as the Senior Editor. The reviewers have opted to remain anonymous.

The reviewers have discussed the reviews with one another and the Reviewing Editor has drafted this decision to help you prepare a revised submission. Because of the extensive textual revisions requested by all reviewers, we refer you to the detailed reviews for specific changes that should be make.

Summary

In this study, Coulcher et al. analyzed the cis-regulatory regions for determinants of dorsal and ventral peripheral nervous system (PNS) in several species of ascidians. Starting with the Phlebobranch Ciona intestinalis, they identified active enhancers for 6 out 7 transcription factor-coding genes that were previously characterized to various extent. They showed that homologous transcription factors have conserved expression patterns in a distant Phlebobranch, *P. mammillata*, and identified corresponding cis-regulatory modules (CRMs) that drive reporter gene expression in the caudal PNS. Using cross-species reporter gene expression assays, between *C. intestinalis* and *P. mammillata*, but also in *Ascidia mentula* and *Phallusia fumigata*, the authors conclude that the identified regulatory DNAs have largely conserved enhancer activities within Phlebobranchs. However, minimal elements showed deprecated activity in cross-species assays, and the Ciona elements also responded differently from endogenous genes in response to perturbations of trans-acting regulators in C. intestinalis.

Next, the authors inspected the spatial and temporal expression patterns of homologous genes in two Stolidobranch species, *Molgula appendiculata* and *Halocynthia roretzi*. They observed a reduced number of caudal epidermal sensory neurons (CESN) in *M. appendiculata* and a restriction to the tip of the tail, which was consistent with restricted patterns of *Nkx*, *Ascl* and *Tox*. Finally, the authors performed CRM swaps between Phlebobranchs and Stolidobranch species, which represent longer evolutionary distances, and showed preliminary evidence that Stolidobranch species interpreted Phlebobranch sequences differently.

Reviewer #1:

This study explored the regulatory mechanisms for the peripheral nervous system of ascidians from an evolutionary viewpoint. The authors of this study performed an impressive suite of laboratory experiments and bioinformatic analyses, covering four ascidian species with more than 250 million years of evolutionary divergence. They report possible conservation of the regulatory mechanism over this evolutionary period, while they revealed some dissimilarities. I found a large value of publishing this study, while I recognized quite a few problematic descriptions of their data, lacking objective and quantitative grounds. I suggest a substantial revision of the manuscript to enhance the objectivity of their arguments based on the points listed below.

1) In many parts of the manuscript, the results are described in too obscure expressions, including “simple”, “conserved”, “divergent”, “typical”, “very few”, “fast”, and “large” that do not accompany quantitative or comparative information. For example, the word “conserved” appears seven times in the Introduction, and they do not provide any logical and testable meaning. Those parts of the manuscript should be carefully rewritten to logically convey the intension.

2) Among vertebrates, we already know quite a few cases of evolutionary conservation of regulatory mechanisms over 300 million years, e.g., between mouse and zebrafish. To me, the scope of this study seems confined to urochordate biology and evolution, but it is advisable to discuss the conservation levels across the ascidian and other lineages along the timeline.

3) The citation in paragraph four of the Introduction should be limited to the references about the divergence time, that is, Delsuc et al., 2018.

4) What ascertains the orthology between genes and genomic regions of different species analyzed in this study? State if not done in the manuscript.

5) The use of the words “player” and “actors” are too abstract and should be refrained.

6) Clarify what “phylogenetic footprints” means.

7) Enrichment of particular genomic regions revealed by ATAC-seq should be called “enrichment”, but not “peak”.

8) Splice isoforms are mentioned only for Tox. Consider mentioning the isoform multiplicity for the other genes, if any.

9) The word “manuscript” in subsection “Expression and regulation of caudal PNS genes” should be replace with “study”.

10) In subsection “Expression and regulation of caudal PNS genes”, is it clear from which species the sequence of the recombinant Bmp4 protein is derived? State which species, if not included in Materials and methods.

11) It was not clear to me what this part means “if Molgula as a divergent species is a valid hypothesis”. The word “divergent” without any quantitative basis does not provide any chance for validating it. This part needs to be elaborated.

Reviewer #2:

Coulcher et al. present an in-depth identification of PNS midline enhancers in divergent ascidians. They use cross-species transgenesis to obtain evidence that enhancer DNAs with very little sequence conservation nonetheless produce similar patterns of gene expression. The study is exhaustive and exhausting and should be of interest to specialists in the field, but the authors will need to rework the presentation in order to engage a broader audience.

A few specific comments follow:

The identification of separate early and late Msx enhancers for expression in the b6.5 lineage is a highlight of the study and should be emphasized in a revised manuscript. The authors should discuss why separate enhancers are used and whether they are likely to interact with different classes of sequence-specific transcription factors.

It is curious that the authors identified dorsal-specific midline enhancers but no ventral-specific enhancers. They do not provide a very compelling explanation for this finding. The ventral midline PNS might be a "throwback" to simpler Deuterostomes that contain both dorsal and ventral nerve cords.

I'm not convinced that the authors have done the most thorough job of identifying TFs that recognize the different enhancers. As a result, it remains an open question whether divergent enhancers use a similar logic to produce similar expression profiles, or if there have been substantive changes in the gene regulatory networks governing the specification of ventral midline in divergent ascidians.

Reviewer #3:

This is a very rich study, reporting on heroic efforts to characterize gene expression patterns, enhancers and their activity across multiple species of ascidians, while focusing on the peripheral nervous system, which is of broad interest. The specific data reported in this manuscript will thus be of interest for both specialists in the field, and outside.

For example, it is interesting to identify distinct ventral vs. dorsal elements for genes like *Msx* and *Ascl.b*, but pan-midline elements for *Tox* and *Klf*, which indeed suggests that the more "upstream" factors integrate distinct regional inputs, while they a common logic controls the core midline factors. However this model needs further support and clarification.

Besides the large amount of specific new information, it is quite intriguing to observe that minimal enhancers are more likely to fail in cross-species assays, and show more discrepancies with endogenous gene expression in experimental conditions that perturb trans-acting factors. These observations suggest that, even when the activity entire cis-regulatory systems is conserved, individual elements are subjected to drift, which is in keeping with the lack of sequence conservation and could be discussed more explicitly in the light of emerging concepts about complex cis-regulatory systems (e.g. so-called shadow enhancers).

Finally, a key merit of the comparison with Stolidobranch is to show evolutionary changes in gene expression patterns (e.g. restriction to the tip of the tail in Molgula, reduced domain of neurogenic competence in Halocynthia), in addition to CRM drift, which has been documented in several other contexts before. The two are difficult to disentangle though, and this last section seems more preliminary; the paper would thus probably benefit from focusing on the sections on Phlebobranchs.

In sum, this is a very rich manuscript with numerous strengths, which would nonetheless benefit from extensive modifications for clarity, a better focus and more in-depth analysis of key points as detailed below.

Major comments

The paper is often very difficult to follow, especially because critical information regarding endogenous and reporter gene expression, ATAC-seq peaks, deletions and reporter activity are scattered across different figures, especially when cross-species comparisons are evoked. Specific cases are listed below:

Figure 1F: difficult to relate the deletions to the peaks of conservation and accessibility if they are not aligned.

Indicate clearly (in main text and figure) what is the core promoter used.

The arrangement of figures makes it very difficult to follow. E.g. in Supplementary file 2, *Klf* comes after Nkx-tun3 but it comes before in the text.

Figure 3—figure supplement 2: *Klf* very hard to connect the pattern, with the ATAC-seq data, with the deletions and the scoring. There should be a way to better integrate these informations and make the figures easier to follow.

Nomenclature needs to be consistent across text and figures, especially for gene names (Tox vs CAGF, Dll.C vs. Dlx.c, etc…). Use published guidelines, and for TFs, the names used in the latest version of the genome.

Not clear from Figure 5 and supplement what the authors mean by "The topology of early and late *Msx* CRMs is strikingly similar between *C. intestinalis* and *P. mammillata*".

Main conclusion on "strong similarities" would be better illustrated by placing the diagram representing genes and CRMs of *C. intestinalis* and *P. mammillata* side-by-side to highlight said similarities, and give the reader a chance to evaluate the differences.

Summary table of stages and tissues stained for each gene between species would be useful to compare patterns across species. And differences could be presented as heatmaps, similar to other panels in the paper (e.g. Figure 7C). Currently the extensive in situ panels are very nice but difficult to compare to each other.

An important aspect of the paper is the relative contribution of individual CRMs to endogenous expression. While there is evidence that genes like *Msx.b* and *Ascl.b* are regulated differently, and part through separate elements in the ventral and dorsal midline, this should still be clarified. Specifically:

The importance of individual elements is never tested using CRISPR/Cas9 mutations followed by in situ hybridization, while this would strongly complement some of the conclusions based exclusively on reporter assays. For no gene is it entirely clear how the identified elements contribute to endogenous regulation, also since there are several other peaks in the regions that were not included in the analysis. E.g. *Dlx.c* had >14 peaks in the ATAC-seq data, out of which one (1) was included in the reporter constructs.

The existence of separate vs. shared CRMs is intriguing and interesting, except this is not as clear for *Ascl.b*, since both CiAshdo4 and CiAshdo5 are still active in the VML, and an intronic accessible region has not been tested. Moreover, Nkx.C is presented as an upstream regulator and absent from the *Msxb* GRN in Roure and Darras, 2016, but here it is incorporated in the core midline PNS network. Since its intron has not been tested, one cannot rule out that there is regulatory activity that would distinguish between ventral and dorsal midlines, which is also true for Dlx.c. This basically casts doubt on the model shown in Figure 4A.

For *Ascl.b*, the Supplementary file 2 makes it look like the intron should be added to the upstream region for thorough testing. In general, one intronic region also appears accessible for Nkx-tun3.

Multiple CRMs in *Msx*, *Klf*, *Nkx* and *Dlx.c* also begs the question as to whether these play redundant roles (like shadow enhancers), or have seemingly overlapping activities in reporter assays, but contribute to different aspects of gene expression.

X-gal staining does not report on the dynamics of enhancer activity: could the two CRMs act at different times? Different levels? Like so-called "shadow" enhancers? Different domains. This needs to be clarified to be useful.

Discussion “ this opens an interesting perspective in identifying and determining the properties of the DNA elements that confer robustness.”: this echoes the growing recognition of integrated cis-regulatory systems with multiple elements contributing to gene regulation.

In general, the authors need to integrate the sequence of elements, including conservation, putative binding sites, and known regulators to paint a more complete and clearer picture of enhancer function in the first part, and of possible conservation -or divergence- of regulatory linkages in the comparative part, especially with regards to the ventral vs. dorsal inputs and the pan-medial/core network for PNS specification.

BMP and Notch signaling regulate ventral MDL and PNS, although crude assays do not allow for refined spatial and temporal resolution, so strictly speaking the time of space at which these regulators act could vary. This is potentially important because the conclusions that will be drawn regarding CRM evolution depend on whether the trans-acting inputs are exactly the same or not.

Sequence analysis, especially in terms of known regulatory linkage within the PNS GRN would strengthen the conclusion on conservation, while clarifying the impact of sequence divergence (i.e. redistribution of binding sites, etc…).

Finally, the analysis of Stolidobranchia CRMs is cursory at this point, and one notes that presumably CRMs for the same genes could be obtained from the sequenced genomes of other tailed Molgula species (e.g. *Molgula oculata* or *Molgula occidentalis*). Considering the density of the current manuscript, the cursory analysis of Stolidobranch vs. Phlebobranch comparisons, and the fact that this has been done before, I would probably recommend focusing on the within Phlebobranch comparisons and strengthening the paper by better integrating the endogenous and reporter expression data with previously established regulatory linkage, sequences and putative TF binding sites. The latter is mentioned as future work in the Discussion, but it is not clear why it cannot be incorporated here, in a focused but clarified version of the study.

Drift between Phlebobranch and Stolidobranch enhancers for conserved gene expression patterns has been described qualitatively before, so the key novelties are the within order comparisons and the evolutionary changes, especially the seemingly reduced neurogenic compartments in Stolidobranchs.

Localized expression of *Tox* and *Ascl.b* in Halocynthia, as opposed to pan-VMDL in Phlebobranch is also a potentially interesting difference (i.e. not the whole midline has neurogenic potential). => possible conservation of the midline program, but drift in its linkage to the neurogenic program in CESNs? This would be very interesting, but difficult to address in a revision.

[Editors' note: further revisions were suggested prior to acceptance, as described below.]

Thank you for submitting your article "Conservation of peripheral nervous system formation mechanisms in divergent ascidian embryos" for consideration by *eLife*. Your article has been reviewed by three peer reviewers, one of whom is a member of our Board of Reviewing Editors, and the evaluation has been overseen by Marianne Bronner as the Senior Editor. The reviewers have opted to remain anonymous.

The reviewers have discussed the reviews with one another and the Reviewing Editor has drafted this decision to help you prepare a revised submission.

Summary:

This study presents an elaborate analysis employing open chromatin data investigation and promoter assays to detect binding sites on a genome-wide scale for several transcription factors by encompassing multiple ascidian species. The study was conducted to elucidate the evolutionary origin of the mechanism governing the formation of the peripheral nervous system (PNS). The results highlighted the unexpected commonalities between those species that are divergent from each other over 200 million years since their split and supported the early origin of the regulatory mechanism for the PNS formation.

Revisions:

The authors took a lot of the main comments to heart in the previous revision, but the manuscript became loaded with details and unnecessarily lengthy. For the sake of clarity of keys in the original data and the conclusion, the manuscript should now be streamlined before making a decision. I advise the authors to achieve this by deleting speculative sentences in Results, for example, “the loss of these sites might thus be responsible of the absence of activity in the VML.” and “and this may explain why we did not observe CRM active only in the VML.” in the *Msx* section This type of information also in the other TF sections should be deleted if not directly linked to the main conclusion, or substantially shortened after moving to Discussion.

---

## [Author Response]

Reviewer #1:This study explored the regulatory mechanisms for the peripheral nervous system of ascidians from an evolutionary viewpoint. The authors of this study performed an impressive suite of laboratory experiments and bioinformatic analyses, covering four ascidian species with more than 250 million years of evolutionary divergence. They report possible conservation of the regulatory mechanism over this evolutionary period, while they revealed some dissimilarities. I found a large value of publishing this study, while I recognized quite a few problematic descriptions of their data, lacking objective and quantitative grounds. I suggest a substantial revision of the manuscript to enhance the objectivity of their arguments based on the points listed below.1) In many parts of the manuscript, the results are described in too obscure expressions, including “simple”, “conserved”, “divergent”, “typical”, “very few”, “fast”, and “large” that do not accompany quantitative or comparative information. For example, the word “conserved” appears seven times in the Introduction, and they do not provide any logical and testable meaning. Those parts of the manuscript should be carefully rewritten to logically convey the intension.

We have revised the text trying to be more precise and specific by bringing additional information and references. We have also added two summary figures (Figure 8—figure supplement 3 and Figure 10) that allow to evaluate the degree of conservation in a visual way.

2) Among vertebrates, we already know quite a few cases of evolutionary conservation of regulatory mechanisms over 300 million years, e.g., between mouse and zebrafish. To me, the scope of this study seems confined to urochordate biology and evolution, but it is advisable to discuss the conservation levels across the ascidian and other lineages along the timeline.

We have added a short reference to comparative studies in vertebrates in the Discussion.

3) The citation in paragraph four of the Introduction should be limited to the references about the divergence time, that is, Delsuc et al., 2018.

This has been corrected.

4) What ascertains the orthology between genes and genomic regions of different species analyzed in this study? State if not done in the manuscript.

To identify orthologous genes, we have performed reciprocal blast and used phylogenies available on the Aniseed database (for *Ciona*, *Phallusia* and *Halocynthia*). For *Molgula appendiculata*, we have only performed reciprocal blast but no proper phylogenetic analysis. This has been added in the Materials and methods section.

In our study, we have not directly adressed the issue of orthologous genomic regions. We have guided our choice for CRM candidates by DNA sequence conservation in non-coding regions. These “phylogenetic footprints” could be obtained from sister species (*Ciona intestinalis* and *Ciona savignyi*; *Phallusia mammillata* and *Phallusia fumigata*). However, non-coding DNA does not align between *Ciona* and *Phallusia*. This is one of the difficulty and interest of comparative CRM studies in ascidians. We have discussed this aspect in the text.

5) The use of the words “player” and “actors” are too abstract and should be refrained.

We have used more specific terms in all instances.

6) Clarify what “phylogenetic footprints” means.

Done.

7) Enrichment of particular genomic regions revealed by ATAC-seq should be called “enrichment”, but not “peak”.

This has been corrected.

8) Splice isoforms are mentioned only for Tox. Consider mentioning the isoform multiplicity for the other genes, if any.

*Tox* is the only case with evidence for isoforms.

9) The word “manuscript” in subsection “Expression and regulation of caudal PNS genes” should be replace with “study”.

Done.

10) In subsection “Expression and regulation of caudal PNS genes”, is it clear from which species the sequence of the recombinant Bmp4 protein is derived? State which species, if not included in Materials and methods.

We have used recombinant Bmp4 protein from zebrafish. This was included in the Material and Methods section.

11) It was not clear to me what this part means “if Molgula as a divergent species is a valid hypothesis”. The word “divergent” without any quantitative basis does not provide any chance for validating it. This part needs to be elaborated.

We are sorry of the confusion, we have modified the text. The divergent status of *Molgula* was developed in a previous paragraph.

Reviewer #2:Coulcher et al. present an in-depth identification of PNS midline enhancers in divergent ascidians. They use cross-species transgenesis to obtain evidence that enhancer DNAs with very little sequence conservation nonetheless produce similar patterns of gene expression. The study is exhaustive and exhausting and should be of interest to specialists in the field, but the authors will need to rework the presentation in order to engage a broader audience.A few specific comments follow:The identification of separate early and late Msx enhancers for expression in the b6.5 lineage is a highlight of the study and should be emphasized in a revised manuscript. The authors should discuss why separate enhancers are used and whether they are likely to interact with different classes of sequence-specific transcription factors.

Following the reviewer's suggestion, we have examined TFBSs for the active CRMs that we have identified in this study (see below). An interesting hypothesis (schematized in Figure 1—figure supplement 2B) is as follows. The early/proximal *Msx* CRM is activated following neural induction and allows expression in DML precursors. The late/distal CRM would initiate *Msx* expression in DML at gastrula/neurula stages by Msx itself.

It is curious that the authors identified dorsal-specific midline enhancers but no ventral-specific enhancers. They do not provide a very compelling explanation for this finding. The ventral midline PNS might be a "throwback" to simpler Deuterostomes that contain both dorsal and ventral nerve cords.

We were also disappointed of not finding ventral-specific enhancers for *Msx* and *Ascl.b* through rather extensive deletion analyses. We had proposed in the first version of the manuscript that ventral and dorsal CRMs were overlapping. Following the reviewer's suggestion for TFBS identification (see below), we now show in Figure 1—figure supplement 2 that candidate sites for ventral TFs are more abundant than sites for dorsal TFs, and that both types of sites are intermingled or even overlapping. It is thus possible that “by chance” we have eliminated ventral TFBS and revealed dorsal-specific activity, while other deletions have eliminated both dorsal and ventral TFBS. Site-specific mutagenesis will be required to test this hypothesis.

Regarding the evolutionary interpretation of the ventral *vs* dorsal PNS, we feel it is still difficult comparing ascidian tail PNS with nervous systems from echinoderms and hemichordates. However, the existence of a Bmp-induced ventral neurogenic territory is evident in both ascidians and amphioxus, suggesting an ancestral origin at least in chordates. Since this territory does not exist in vertebrates, it is likely that it has been lost. In our group, we are currently working on a comparative project on ascidians and amphioxus to test the hypothesis that this ancestral ventral program has been co-opted at the dorsal neural plate border and has led to the emergence of vertebrate neural crest and placodes. An attractive hypothesis would be that sites for dorsal TFs (*Msx* for example) have appeared in an “ancestral” ventral neurogenic epidermis enhancer. However, the results that we present in the current study do not bring results strong enough to discuss this evolutionary aspect.

I'm not convinced that the authors have done the most thorough job of identifying TFs that recognize the different enhancers. As a result, it remains an open question whether divergent enhancers use a similar logic to produce similar expression profiles, or if there have been substantive changes in the gene regulatory networks governing the specification of ventral midline in divergent ascidians.

We also thought that comparing TFBS composition of enhancers with conserved activity but originating from different species would be straightforward in determining whether the regulatory logic is conserved or not. We did not show any data on this aspect in the first version because of the following weaknesses:

– the caudal PNS GRN is still poor, mainly built from expression data, overexpression of midline TFs and loss-of-function of a few genes. Hence, direct regulations and the identity of TFs that are actually involved are not known.

– gene regulation is modular with redundant, shadow or distributed enhancers. Given that DNA sequence conservation between ascidian species from different genera (*Ciona intestinalis* and *Phallusia mammillata* for example) is virtually absent, it is not possible to define orthologous CRMs. While topology, sequence conservation (between closely related species: *Ciona intestinalis* and *Ciona savignyi*) and accessibility (ATAC-seq) could be used to propose orthologous CRMS; we are not sure that we will be comparing equivalent functional units.

We nevertheless did our best to address this question:

– we have gathered available information and proposed some hypotheses for gene regulation to improve the VDML GRN (Figure 1—figure supplement 1A and Materials and methods)

– we have gathered TF binding specificities (Figure 1—figure supplement 1B) and mapped TFBS on CRMs aligned with their counterpart from closely related species (*C. intestinalis* and *C. savignyi* alignment on one side, and *P. mammillata* and *P. fumigata* alignment on the other side). We have restricted our analysis to CRMs smaller than 1 kb and to TFBS that would be conserved in the alignments (assuming that regulatory logic is conserved for close species).

This analysis in *Ciona intestinalis* allowed us to:

– propose some direct regulations for a number of genes

– suggest that each gene of the VDML network is regulated differently (in particular, the early genes *Msx* and *Ascl.b* do not seem to be regulated similarly)

– propose that unidentified TFs are likely to play important roles

Unfortunately, given that our in vivo analysis of CRMs is not exhaustive, we could make comparisons between *Ciona* and *Phallusia* for only 2 genes. However, this gave support for a similar logic in both species: *Msx* (direct activation by Bmp signaling in the VML and by *Msx* in the DML) and *Tox* (direct activation by *Ascl.b*).

Reviewer #3:[…]Major commentsThe paper is often very difficult to follow, especially because critical information regarding endogenous and reporter gene expression, ATAC-seq peaks, deletions and reporter activity are scattered across different figures, especially when cross-species comparisons are evoked. Specific cases are listed below:

We acknowledge that the rather large amount of data that we present may make the reading difficult. We had tried to present the actual data without simplification, while trying at the same time to keep systematic graphical representations for clarity. We have now followed reviewer's advice and made several changes that are described below. We have also added two new summary figures (Figure 8—figure supplement 3 and Figure 10) that schematize the comparative part. We also have added new analyses (TFBS, see below) that we hope will not make the paper even more confusing.

Figure 1F: difficult to relate the deletions to the peaks of conservation and accessibility if they are not aligned.

Conservation and accessibility have now been reported in all schemes describing various CRMs and their activity.

Indicate clearly (in main text and figure) what is the core promoter used.

In all cases, except 3 CRMs (Ciinte.Nkxtun3-upstream, Ciinte.Ascl.b-upstream and Ciinte.Bhlhtun1-upstream), the core promoter is the one of *Ciinte.Fog* (Rothbacher et al., 2007). This information can be found in Materials and methods and Supplementary file 1. We have now added a sentence at the beginning of the Results section. We have chosen not to repeat this redundant information in every figure legend.

The arrangement of figures makes it very difficult to follow. E.g. in Supplementary file 2, Klf comes after Nkx-tun3 but it comes before in the text.

This has been fixed.

Figure 3—figure supplement 2: Klf very hard to connect the pattern, with the ATAC-seq data, with the deletions and the scoring. There should be a way to better integrate these information and make the figures easier to follow.

As mentioned above, accessibility and conservation are now shown directly below the schemes of the various tested CRMs. We hope figures are clearer and easier to follow.

Nomenclature needs to be consistent across text and figures, especially for gene names (Tox vs CAGF, Dll.C vs. Dlx.c, etc…). Use published guidelines, and for TFs, the names used in the latest version of the genome.

Gene and CRM names have been modified throughout text and figures to adhere to the proposed nomenclature (Stolfi et al., 2015).

Not clear from Figure 5 and supplement what the authors mean by "The topology of early and late Msx CRMs is strikingly similar between *C. intestinalis* and *P. mammillata*".

We have clarified this point.

Main conclusion on "strong similarities" would be better illustrated by placing the diagram representing genes and CRMs of *C. intestinalis* and *P. mammillata* side-by-side to highlight said similarities, and give the reader a chance to evaluate the differences.

We have moderated our conclusions. The idea that regulatory landscapes were similar emerged from the position of the CRMs active in VDML relative to the gene of interest, and from the *Msx* locus. While the *Msx* case is striking, the position of CRMs for other genes (within 3 kb upstream) might correspond to real similarities, but might be simply due to the fact that this is where we looked for them in the first place. Since other VDML CRMs might exist (see below), we removed this idea of “regulatory landscape”.

Summary table of stages and tissues stained for each gene between species would be useful to compare patterns across species. And differences could be presented as heatmaps, similar to other panels in the paper (e.g. Figure 7C). Currently the extensive in situ panels are very nice but difficult to compare to each other.

We have added a supplemental figure to Figure 8 (Figure 8—figure supplement 3) that schematizes the expression patterns in the four species.

An important aspect of the paper is the relative contribution of individual CRMs to endogenous expression. While there is evidence that genes like Msx.b and Ascl.b are regulated differently, and part through separate elements in the ventral and dorsal midline, this should still be clarified.

We do agree with the reviewer that transcriptional regulation is complex and involves multiple CRMs that may or may not have similar activity in time and space. But we did not claim that we have exhaustively analyzed the CRM landscape for PNS expression of our genes of interest. The reviewer has to keep in mind that resources or tools to explore CRMs is still limited in ascidians: ChIP-seq and ATAC-seq are just beginning to be used, others techniques like Chromosome Conformation Capture are not yet established. Very little was known for gene regulation and *cis*-regulation of the genes we studied; and we have only used phylogenetic footprints and in vivo transcriptional assays. We have nevertheless reached interesting conclusions that together with cross-species tests bring new hypotheses.

The importance of individual elements is never tested using CRISPR/Cas9 mutations followed by in situ hybridization, while this would strongly complement some of the conclusions based exclusively on reporter assays.

This is obviously the next step that we would like to reach. However, CRISPR/Cas9 has proven very frustrating in our hands, possibly because a higher polymorphism rate in *C. intestinalis* compared to *C. robusta*. In addition, we have so far reached at best 30% mutagenesis rate using electroporation. We also feel that such experiments would be part of a future study.

For no gene is it entirely clear how the identified elements contribute to endogenous regulation, also since there are several other peaks in the regions that were not included in the analysis. E.g. Dlx.c had >14 peaks in the ATAC-seq data, out of which one (1) was included in the reporter constructs.

As discussed below, currently available ATAC-seq data (whole embryos up to neurula stages) may not actually fit our needs: some important regions may not be apparent, and at the same time regions with visible enrichment may not correspond to VDML expression. Tissue-specific ATAC-seq or 3C data should prove better staring points for an extensive analysis of candidate CRMs.

The existence of separate vs. shared CRMs is intriguing and interesting, except this is not as clear for Ascl.b, since both CiAshdo4 and CiAshdo5 are still active in the VML, and an intronic accessible region has not been tested. Moreover, Nkx.C is presented as an upstream regulator and absent from the Msxb GRN in Roure and Darras, 2016, but here it is incorporated in the core midline PNS network. Since its intron has not been tested, one cannot rule out that there is regulatory activity that would distinguish between ventral and dorsal midlines, which is also true for Dlx.c. This basically casts doubt on the model shown in Figure 4A.For Ascl.b, the Supplementary file 2 makes it look like the intron should be added to the upstream region for thorough testing. In general, one intronic region also appears accessible for Nkx-tun3.

We agree that the situation for *Ascl.b* is not as sharp as *Msx* since ventral activity is strongly reduced but not abolished for do4 and do5 CRMs. Yet we have not observed such a situation for the other four genes that we have analyzed and that are expressed later. Of course, we cannot rule out that unidentified CRMs with pan-midline regulation exist for *Msx* and *Ascl.b* and that unidentified dorsal CRMs and ventral CRMs also exist for the other midline TFs. We thus believe that deeper understanding of midline TFs transcriptional regulation will require a substantial amount of work with much finer resolution (in time and space with quantitative approaches) involving much heavier experimental approaches. For example, we propose that the DML expression at gastrula/neurula stage of *Ciinte.Msx* is initiated by Msx itself through the distal CRM (see below). Once *Msx* expression has been launched in both dorsal and ventral midlines, it is possible that the same element is used for maintenance through positive autoregulation. Thus the same genomic element would be at the same time a DML enhancer and a “pan-midline” enhancer.

Regarding the *Ciinte.Ascl.b* gene structure, we have fully sequenced the cDNA clone cien82323; and this indicated that the intron was wrongly predicted in the transcript models (shown in the modified Supplementary file 2). That is why we have selected the CRMs as shown. Yet this clone might be an artifact since it is not well supported by RNA-seq data. Nevertheless, a genomic region covering this region had been previously tested in transcriptional assay and was not active in the tail epidermis (https://www.aniseed.cnrs.fr/aniseed/cisreg/show_cisreg?feature_id=11170652).

The network in Roure and Darras, 2016 is based on overexpression and lacks information on direct interactions. Several structures were actually proposed with *Nkxtun3* (previously *Nkx-C*) included in most.

Multiple CRMs in Msx, Klf, Nkx and Dlx.c also begs the question as to whether these play redundant roles (like shadow enhancers), or have seemingly overlapping activities in reporter assays, but contribute to different aspects of gene expression.X-gal staining does not report on the dynamics of enhancer activity: could the two CRMs act at different times? Different levels? Like so-called "shadow" enhancers? Different domains. This needs to be clarified to be useful.Discussion “ this opens an interesting perspective in identifying and determining the properties of the DNA elements that confer robustness.”: this echoes the growing recognition of integrated cis-regulatory systems with multiple elements contributing to gene regulation.

The contribution of several CRMs with seemingly similar activity is indeed a current trend in *cis*-regulation studies. The role of “redundant” CRMs has been first linked to robustness, their specific roles being revealed following genetic or environmental challenges. Quantitative approaches are now revealing that similar CRMs are intrinsically different. In ascidians, we are aware of only three studies reporting “shadow”/distributed enhancers (Farley et al., 2016; Madgwick et al., 2019; Harder et al., https://doi.org/10.1101/2020.08.07.242016). It seems that we have identified additional ones. However, their analysis would be a separate study.

In general, the authors need to integrate the sequence of elements, including conservation, putative binding sites, and known regulators to paint a more complete and clearer picture of enhancer function in the first part, and of possible conservation -or divergence- of regulatory linkages in the comparative part, especially with regards to the ventral vs. dorsal inputs and the pan-medial/core network for PNS specification.BMP and Notch signaling regulate ventral MDL and PNS, although crude assays do not allow for refined spatial and temporal resolution, so strictly speaking the time of space at which these regulators act could vary. This is potentially important because the conclusions that will be drawn regarding CRM evolution depend on whether the trans-acting inputs are exactly the same or not.Sequence analysis, especially in terms of known regulatory linkage within the PNS GRN would strengthen the conclusion on conservation, while clarifying the impact of sequence divergence (i.e. redistribution of binding sites, etc…).

Since this aspect was also suggested by reviewer 2, we have undertaken a careful analysis of VDML CRMs regarding binding site composition for putative regulators of PNS specification (see above). We would like to stress that we have not come to a definitive conclusion in terms of conservation of regulatory logic. The main reasons are 1) uncertainties of the GRN given the small number of validated direct interactions between a TF and a target gene, and the likely existence of unknown regulators, and 2) the impossibility to define homology relationships between CRMs from *Ciona* and *Phallusia* since non-coding DNA regions do not align.

Finally, the analysis of Stolidobranchia CRMs is cursory at this point, and one notes that presumably CRMs for the same genes could be obtained from the sequenced genomes of other tailed Molgula species (e.g. Molgula oculata or Molgula occidentalis). Considering the density of the current manuscript, the cursory analysis of Stolidobranch vs. Phlebobranch comparisons, and the fact that this has been done before, I would probably recommend focusing on the within Phlebobranch comparisons and strengthening the paper by better integrating the endogenous and reporter expression data with previously established regulatory linkage, sequences and putative TF binding sites. The latter is mentioned as future work in the Discussion, but it is not clear why it cannot be incorporated here, in a focused but clarified version of the study.Drift between Phlebobranch and Stolidobranch enhancers for conserved gene expression patterns has been described qualitatively before, so the key novelties are the within order comparisons and the evolutionary changes, especially the seemingly reduced neurogenic compartments in Stolidobranchs.

We acknowledge that comparison between Phlebobranchs and Stolidobranchs is not as deep as the *Ciona*/*Phallusia* part. However, while comparative approaches have been reported for single genes between *C. intestinalis* and *H. roretzi* (Oda-Ishii et al., 2005; Matsumoto et al., 2008; Takahashi et al., 2009), or for multiple genes between *C. intestinalis* and *Molgula* species (Stolfi et al., 2014), we focus here on multiple genes in multiple species. Importantly, our results suggest that, contrary to some of the above reports (comparing possibly non-equivalent CRMs or using unusually divergent species), regulatory mechanisms might be broadly conserved in ascidians despite extensive drift. We thus consider that this part of the study is of strong interest.

Localized expression of Tox and Ascl.b in Halocynthia, as opposed to pan-VMDL in Phlebobranch is also a potentially interesting difference (i.e. not the whole midline has neurogenic potential). => possible conservation of the midline program, but drift in its linkage to the neurogenic program in CESNs? This would be very interesting, but difficult to address in a revision.

The reviewer suggests an interesting possibility. Another scenario could be “parallel” GRNs in the caudal PNS (Pan-midline: *Msx*->*Klf1/2/4*->*Nkxtun3*->*Dlx.c* and posterior midline: *Ascl.b*-> *Tox*) that would be both required for acquiring a neurogenic potential. In *H. roretzi*, it has been shown that activating Notch inhibits PNS neuron formation (Akanuma et al., 2002). However, Notch inhibition has not been performed, and it is not known whether the entire midline is neurogenic. Such an experiment together with midline TFs loss-of-function would be required to further explore these hypotheses.

[Editors' note: further revisions were suggested prior to acceptance, as described below.]

Revisions:The authors took a lot of the main comments to heart in the previous revision, but the manuscript became loaded with details and unnecessarily lengthy. For the sake of clarity of keys in the original data and the conclusion, the manuscript should now be streamlined before making a decision. I advise the authors to achieve this by deleting speculative sentences in Results, for example, “the loss of these sites might thus be responsible of the absence of activity in the VML.” and “and this may explain why we did not observe CRM active only in the VML.” in the Msx section This type of information also in the other TF sections should be deleted if not directly linked to the main conclusion, or substantially shortened after moving to Discussion.

Following the suggestion of two reviewers we had included an extensive analysis of transcription factor binding sites in the revised manuscript. Unfortunately, adding more information in an already dense manuscript led to a study even more difficult to follow. Following your suggestion, we have kept the description of this aspect minimal in the main text. We nevertheless believe that this new aspect of the study has brought interesting insights, and we have not simply deleted the data but transferred the details to the supplemental figure legends.